# I-LLM: Efficient Integer-Only Inference for Fully-Quantized Low-Bit Large Language Model

## Abstract

Post-training quantization (PTQ) serves as a potent technique to accelerate the inference of large language models (LLMs). Nonetheless, existing works still necessitate a considerable number of floating-point (FP) operations during inference, including additional quantization and de-quantization, as well as non-linear operators such as RMSNorm and Softmax. This limitation hinders the deployment of LLMs on the edge and cloud devices. In this paper, we identify the primary obstacle to integer-only quantization for LLMs lies in the large fluctuation of activations across channels and tokens in both linear and non-linear operations. To address this issue, we propose I-LLM, a novel integer-only fully-quantized PTQ framework tailored for LLMs. Specifically, (1) we develop Fully-Smooth Block-Reconstruction (FSBR) to aggressively smooth inter-channel variations of all activations and weights. (2) to alleviate degradation caused by inter-token variations, we introduce a novel approach called Dynamic Integer-only MatMul (DI-MatMul). This method enables dynamic quantization in full-integer matrix multiplication by dynamically quantizing the input and outputs with integer-only operations. (3) we design DI-ClippedSoftmax, DI-Exp, and DI-Normalization, which utilize bit shift to execute non-linear operators efficiently while maintaining accuracy. The experiment shows that our I-LLM achieves comparable accuracy to the FP baseline and outperforms non-integer quantization methods. For example, I-LLM can operate at W4A4 with negligible loss of accuracy. To our knowledge, we are the first to bridge the gap between integer-only quantization and LLMs. We've published our code on anonymous.4open.science, aiming to contribute to the advancement of this field.

## 1 Introduction

Large Language Models (LLMs) have paved the way for general artificial intelligence with their remarkable performance across a wide range of tasks. However, the rising number of parameters and computing power requirements of LLMs pose significant challenges when it comes to deployment.

Post-training quantization (PTQ) is a powerful technique employed to accelerate the inference process of LLMs. Previous PTQ methods for LLMs have primarily relied on simulated quantization (aka. fake quantization) (Xiao et al., 2022; Shao et al., 2023; Wei et al., 2022b; Yuan et al., 2023a), where integer values are typically used for inputs/outputs and compute-intensive operations are performed using dequantized floating-point (FP) values (as shown in Fig1). Although this scheme offers benefits in scenarios where data transmission bandwidth is limited, it does not effectively reduce computational costs and thus has little effect on compute-bound situations. Besides, non-linear operations (e.g., Softmax and RMSNorm) often involve complex operations, including transcendental functions such as exponential functions and square root functions. These functions are typically performed on dedicated FP units and may require multiple iterations for accurate computation.

In contrast, integer-only quantization (Jacob et al., 2018; Wu et al., 2020; Qin et al., 2022; Li & Gu, 2023; Lin et al., 2022) utilizes low-precision integer arithmetic for all operations, including linear operations (e.g., matrix multiplication) and non-linear operations. It enables quantized models to take full advantage of fast and efficient integer arithmetic units, resulting in promising speedup effects and reduction of latency and power consumption. Additionally, integer-only quantization facilitates

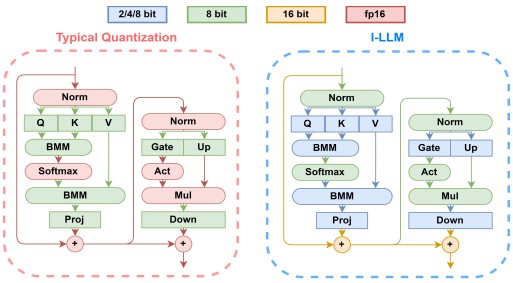 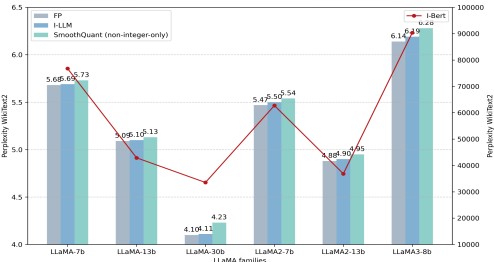

Figure 1: Typical LLM quantization vs. I-LLM. The former requires dequantization and involves FP arithmetic, while the latter performs the entire inference using integer-only arithmetic.

Figure 2: PPL↓ of different PTQ methods on LLaMA family using W8A8. Notably, due to the exceptionally high PPL of I-Bert, a dedicated y-axis has been allocated for its representation.

deployment on popular edge processors specifically designed for embedded, mobile, or IoT devices, which often lack dedicated floating point units. Examples of such edge processors include ARM Cortex-M (WIKIPEDIA, 2024), GreenWaves GAP-9 (Flamand et al., 2018), and Google's Edge TPU (Google, 2024). Note that Turing Tensor Cores in GPU server-class devices also have introduced support for integer logical units, offering notably lower latency compared to FP arithmetic.

However, the aforementioned integer-only methods are designed specifically for CNNs or small Transformer networks (e.g., Bert Kim et al. (2021) and ViT Dosovitskiy et al. (2020)), which renders them inadequate for LLMs. First, they are incapable of straightforwardly supporting the non-linear operators inherent in LLMs, such as SwiGLU and RMSNorm. Furthermore, their accuracy on LLMs deteriorates significantly (as depicted in Figure 2), even when those non-supported operators are executed in full-precision mode. This is because as the model size increasing, the presence of activation outliers in both linear and non-linear layers becomes prominent. As illustrated in Fig 3, Llama2-7B exhibits substantial variations in activation magnitude at both the token and channel levels, making previous approaches ineffective. Last but not least, these methods are limited to 8-bit quantization, whereas LLMs would benefit from lower bit-width quantization (e.g., 4-bit) to address the extensive computational requirements and storage demands. **Consequently, how to accurately perform LLMs with efficient integer-only arithmetic remains an unresolved issue that requires further investigation.**

In this paper, we identify the primary obstacle to integer-only quantization for LLMs lies in the large fluctuation of activations across channels and tokens in both linear and non-linear operators. To address this issue, we introduce I-LLM, a novel integer-only PTQ framework tailored for LLMs: (1) We propose Fully-Smooth Block-Reconstruction (FSBR) to harmonize the variance in activation across channels. While Omniquant Shao et al. (2023) and Smoothquant Xiao et al. (2022) share some similarities, they primarily focus on smoothing the activation in serial norm-linear and parallel linear-linear operations. We argue that mitigating the disparities of all suitable activation-activation and activation-weight pairs of LLMs (see Fig.5) significantly enhances accuracy. For instance, the input of SwiGLU encounters numerous disparities on both token-wise and channel-wise dimensions, as depicted in Fig.4-a. To achieve smoothing on such a non-linear operator, we decom-

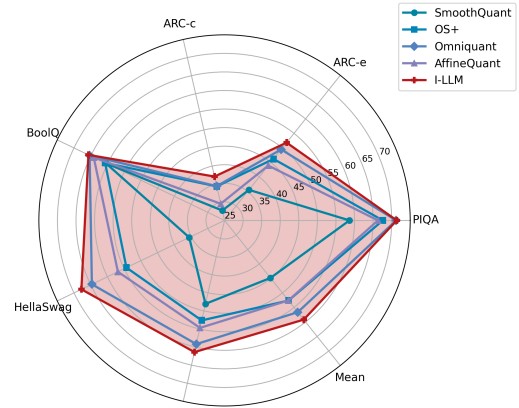

The performance of different quantization methods on six zero-shot datasets using the LLaMA-30B under W4A4 setting.

pose SiLU into $x \cdot \sigma(x)$ to apply FSBR, and as a result the input becomes more consistent and amenable to quantization, as shown in Fig.4-b. (2) To alleviate the degradation resulting from inter-token variations, we present a novel approach called Dynamic Integer-only MatMul (DI-MatMul). DI-MatMul facilitates quantization on full-integer matrix multiplication by employing integer-only operations to quantize the input and outputs dynamically. Traditional static quantization methods, which are characterized by their lack of robustness and adaptability, often falter when encountering input beyond the calibration set. In contrast, DI-MatMul is designed to proactively recognize and

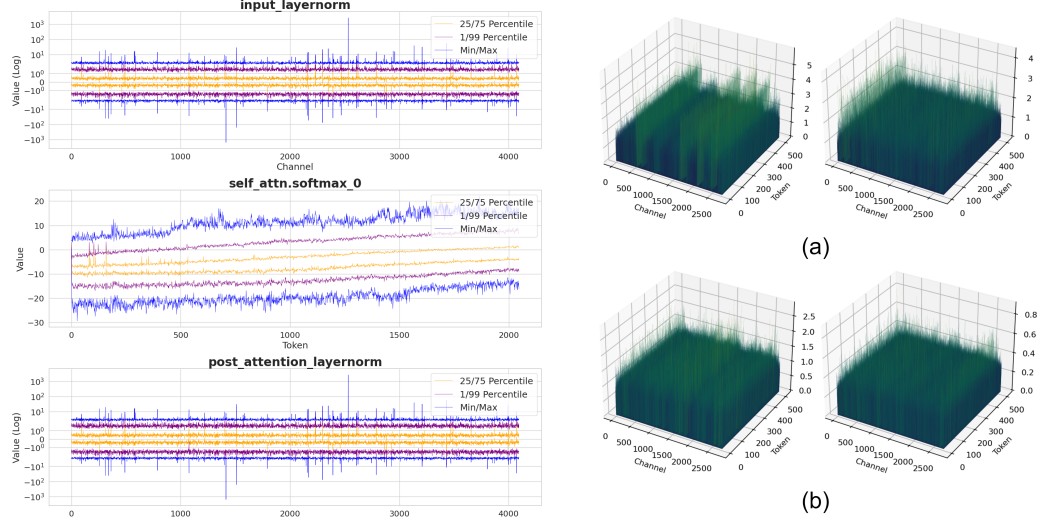

Figure 3: Differences in activations of the non-linear operator in LLaMA2 7b model across the channel/token dimensions.

Figure 4: The input activation distribution of the SwiGLU before FSBR (a) and after FSBR (b).

adapt to the diverse range of input data, thereby reducing quantization errors and enhancing overall model performance. (3) For non-linear operators, we design DI-ClippedSoftmax, DI-Exp, and DI-Norm, which leverage the efficiency of bit shifting to replace complex math calculations while maintaining accuracy. Our contributions are summarized as:

1. We identify the primary obstacle to integer-only quantization for LLMs lies in the large fluctuation of activations across channels and tokens in both linear and non-linear operators. To address inter-channel variations, we propose FSBR to effectively reduces disparities among all suitable activation-activation and activation-weight pairs, thereby markedly improving accuracy.

2. We attribute the failure of previous integer-only quantization methods on LLMs to various range in activation tokens. To tackle this problem, we introduce DI-MatMul, which enables dynamic quantization on input and output through full-integer matrix multiplication. DI-MatMul proactively adapts to the range of input, minimizing quantization errors and improving overall performance.

3. We introduce DI-Exp, DI-ClippedSoftmax, and DI-Norm, innovative integer-only operators that harness the power of bit shifting to replace complex mathematical computations within the non-linear functions of LLMs, without compromising on accuracy.

4. To the best of our knowledge, this work represents the first attempt to utilize integer-only quantization for LLMs, enabling their deployment on edge devices that lack floating-point capabilities. Experiments demonstrate remarkable accuracy when compared to SOTA non-integer quantization techniques; for instance, I-LLM on LLAMA-13b achieves an approximate 20% reduction in perplexity. Additionally, our integer-only algorithm achieves a 2x improvement in computational performance and a 3.5x reduction in memory usage compared to the full-precision model.

## 2 RELATED WORK

**LLMs Quantization.** LLMs quantization can be broadly categorized into weight-only and weight-activation quantization. To alleviate the computational burdens, some works (Liu et al., 2023b; Frantar et al., 2022; Chee et al., 2023; Lin et al., 2023; Lee et al., 2023; Shang et al., 2023; Kim et al., 2023; Dettmers et al., 2023) make efforts in weight-only quantization. GPTQ (Frantar et al., 2022) and QuIP (Chee et al., 2023) achieve high compression rates by optimizing matrix multiplications operation. AWQ (Lin et al., 2023) and OWQ (Lee et al., 2023) demonstrate performance improvements by accounting for the impact of activation outliers on weight quantization. Moreover, works such as QLORA (Dettmers et al., 2024), QA-lora (Xu et al., 2023) and LoftQ (Li et al., 2023b) leverage Parameter Efficient Fine-Tuning (PEFT) techniques to achieve weight compression with fine-tuning tasks. Different with weight-only quantization methods, towards to accelerate the LLMs inference, the weight-activation quantization methods (Wei et al., 2022b; Yuan et al., 2023a; Wei et al., 2023; Li et al., 2023a; Yuan et al., 2023b; Yue et al., 2024) quantize both the weights and activations, including the KV cache. The primary challenge in quantizing activations lies in

outliers, leading to significant quantization errors. To tackle this issue, ZeroQuant (Yao et al., 2022) proposes a fine-grained hardware-friendly quantization scheme for both weight and activations. SmoothQuant (Xiao et al., 2022) migrates the quantization difficulty from activations to weights with a mathematically equivalent transformation (i.e., per-channel scaling). OmniQuant (Shao et al., 2023) further enhances performance by training the quantization parameters. While these methods have mitigated the quantization error, their inference pipelines still involve partially FP operations on non-linear operators such as Softmax, Normalization, and SiLU. In this study, our focus on achieving Integer-only inference for LLMs model using advanced PTQ (Li et al., 2021; Wei et al., 2022a; Liu et al., 2023a; Zhou et al., 2024) techniques.

**Integer-only Quantization.** Current quantization methods for LLMs often involve dequantized FP operations during inference, limiting the utilization of efficient low-precision arithmetic units. Integer-only quantization, eliminating dequantization, enables complete inference using Integer-only arithmetic, promising enhanced model acceleration. Previous approaches (Jacob et al., 2018; Yao et al., 2021) leverage dyadic arithmetic for Integer-only pipeline on CNNs, but these are tailored for linear operations and are unsuitable for non-linear operations in ViTs. Applying INT arithmetic solely to linear operations while retaining FP arithmetic for non-linear ones maybe a straightforward solution, but this demands custom hardware and introduces computational overheads. Advancements include Fully-8bit (Lin et al., 2021) and I-BERT (Kim et al., 2021), which address non-linear operations through employs L1 LayerNorm and INT polynomial approximations for the non-linear operations. However, these methods face inefficiencies or fail to fully exploit hardware advantages. Based on I-BERT (Kim et al., 2021), FQ-ViT (Lin et al., 2022) extends INT arithmetic to part of the operations but overlooks significant non-linear operations like GELU. While some methods (Stevens et al., 2021; Zhu et al., 2020) retain FP arithmetic during approximation, they cannot meet the requirement for Integer-only arithmetic. Recently, I-ViT (Li & Gu, 2023) introduces Integer-only quantization for ViTs, yet its suitability for LLMs is questionable due to differing data distributions, and its computational graph includes partially INT32 precision operations. In this paper, we focus on Integer-only inference for LLMs, maintaining the entire computational graph at INT8 precision or lower bit (e.g., 6/4-bit), enhancing LLMs' inference efficiency on edge processors.

## 3 METHOD

**Challenges of Integer-Only Quantization for Large Language Models.** Presently, integer-only LLMs encounter two primary hurdles: (1) quantizing the activation of LLMs, especially those originating from non-linear operators, poses a formidable challenge. As evidenced in Fig 3, the divergence in these non-linear activations often surpasses that of linear operators, particularly pronounced in models such as LLaMA (Touvron et al., 2023a). Previous methods have failed to address these non-linear activations, and straightforwardly quantizing non-linear layers may lead to substantial accuracy degradation. (2) prior integer-only quantization techniques have overlooked the distinctive traits of LLMs, including divergent activation scales and the substantial overhead of loading large-scale weights. Even the W8A8 method introduced in I-BERT can lead to catastrophic outcomes, as shown in Fig 2, let alone more aggressive quantization methods like W6A6 or W4A4.

In this section, we introduce a novel integer-only quantization framework termed I-LLM. As illustrated in Fig 5, this framework incorporates a differentiable approach within the Post-Training Quantization (PTQ) paradigm, termed Fully-Smooth Block Reconstruction. This method is designed to effectively balance all feasible parameter pairs, as elaborated in Section 3.2. Furthermore, we advance the develop of dynamic quantization within the integer-only context by introducing DI-MatMul, a dynamic integer-only matrix multiplication method, which is elucidated in Section 3.3. Additionally, we detail integer-only non-linear approximations, including DI-ClippedSoftmax, DI-exp, and DI-Norm, that are built upon DI-MatMul and are presented in Section 3.4. These operators facilitate 8-bit input activations while minimizing accuracy loss.

### 3.1 BASIC MATHEMATICAL NOTATIONS

Matrices are denoted by bold uppercase letters such as $X$, vectors by bold lowercase letters such as $x$. Floating-point and integer numbers are distinguished using the superscript $I$, for example, $x^I$ for integers and $x$ for floating-point numbers. The notation $\mathcal{Q}$ means the quant function, $\lfloor \cdot \rfloor$ represents the floor function, and $\lfloor \cdot \rceil$ denotes rounding to the nearest integer. The symbol $\otimes$ indicates

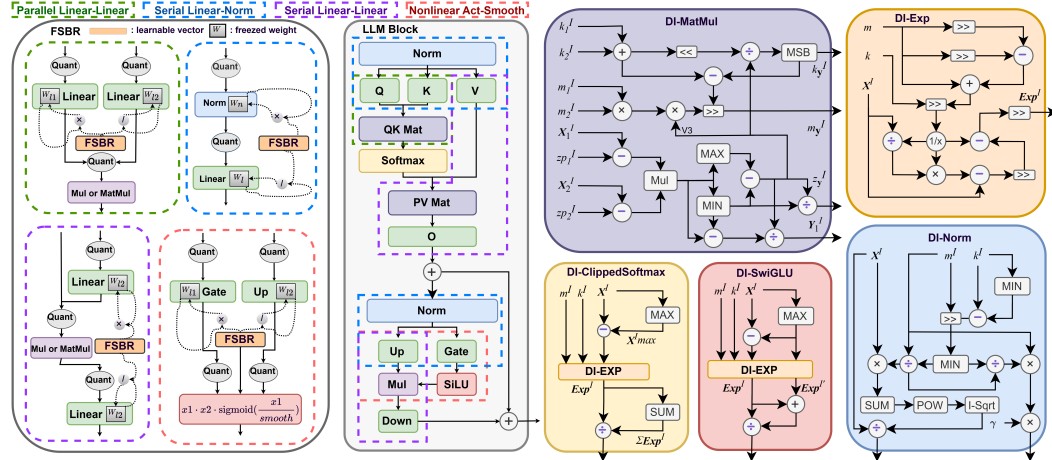

Figure 5: Details of I-LLM in a transformer block. The left side of the figure illustrates various paradigms for channel-wise smoothing during the FSBR process. The right side depicts the integer-only execution pipeline for both linear operators, such as matrix multiplication (MatMul), and non-linear operators.

element-wise multiplication, while $\oslash$ denotes element-wise division. Other mathematical symbols follow their standard definitions.

### 3.2 FULLY-SMOOTH BLOCK-RECONSTRUCTION (FSBR)

As mention above, to mitigate the issue of non-linear layer activations in LLMs being affected by channel and token differences (Fig 3 and 4), we propose Fully-Smooth Block-Reconstruction (FSBR). An intuitive method is to train a smoothing coefficient for all activations and weights to aid in restoring the model's quantization accuracy, especially the quantization accuracy of non-linear operators. Therefore, we consider the activations of all non-linear layers and learn the smoothing coefficients for all possible equivalent smoothing transformations at the channel level. On the left side of Fig 5, four paradigms for achieving inter-channel smoothing during block reconstruction are shown: Parallel Linear-Linear, Serial Linear-Norm, Serial Linear-Linear, and NonLinear Act-Smooth. The first three smoothing methods are indeed linear, making them easier to implement.

However, when addressing non-linear operators (NonLinear Act-Smooth) in LLMs, such as the gated activation function (SwiGLU), it becomes challenging to apply any linear transformations to them. For convenience, we can represent SwiGLU using the following formula:

$$
\begin{aligned}
\mathrm{SwiGLU}(\boldsymbol{x}, \boldsymbol{W}, \boldsymbol{V}, \boldsymbol{b}, \boldsymbol{c}) &= \mathrm{SiLU}(\boldsymbol{x}\boldsymbol{W} + \boldsymbol{b}) \otimes (\boldsymbol{x}\boldsymbol{V} + \boldsymbol{c}) \\
&= \boldsymbol{x1} \otimes \boldsymbol{x2} \otimes \sigma(\boldsymbol{x1})
\end{aligned}
\tag{1}
$$

Where $x \in \mathbb{R}^{ic}$ is a vector, $\boldsymbol{W}, \boldsymbol{V} \in \mathbb{R}^{ic \times oc}$ are weight matrices, and $\boldsymbol{b}, \boldsymbol{c} \in \mathbb{R}^{oc}$ represent biases. $\boldsymbol{x1} = \boldsymbol{x}\boldsymbol{W} + \boldsymbol{b}$, $\boldsymbol{x2} = \boldsymbol{x}\boldsymbol{V} + \boldsymbol{c}$. SiLU stands for Sigmoid-Weighted Linear Unit. $\sigma$ represents the sigmoid activation function. In order to jointly balance the activation and weight quantization difficulties of $\boldsymbol{x1}$ and $\boldsymbol{x2}$, we introduce a smoothing factor $\boldsymbol{s}$, expressed in the formula as follows:

$$
\begin{aligned}
\mathrm{SwiGLU}(\boldsymbol{x}, \boldsymbol{W}, \boldsymbol{V}, \boldsymbol{b}, \boldsymbol{c}) &= (\boldsymbol{x1} \otimes \boldsymbol{s}) \otimes (\boldsymbol{x2} \oslash \boldsymbol{s}) \otimes \sigma'(\boldsymbol{x1}) \\
&= \boldsymbol{x1'} \otimes \boldsymbol{x2'} \otimes \sigma'(\boldsymbol{x1'})
\end{aligned}
\tag{2}
$$

Where $\boldsymbol{W'} = \boldsymbol{W} \otimes \boldsymbol{s}$, $\boldsymbol{b'} = \boldsymbol{b} \otimes \boldsymbol{s}$, $\boldsymbol{V'} = \boldsymbol{V} \oslash \boldsymbol{s}$, $\boldsymbol{c'} = \boldsymbol{c} \oslash \boldsymbol{s}$, $\boldsymbol{x1'} = \boldsymbol{x}\boldsymbol{W'} + \boldsymbol{b'}$, $\boldsymbol{x2'} = \boldsymbol{x}\boldsymbol{V'} + \boldsymbol{c'}$, $\sigma'(\boldsymbol{x1'}) = \sigma(\boldsymbol{x1'} \oslash \boldsymbol{s})$. As shown by the red dashed box in Fig 5, during reconstruction, we first incorporate $\boldsymbol{s}$ into the weights. After calculating $\boldsymbol{x1'}$ and $\boldsymbol{x2'}$, we quantize them before proceeding with the computation. The forward inference process of SwiGLU can be expressed as: $\mathrm{SwiGLU}(\boldsymbol{x}) = \mathcal{Q}(\mathcal{Q}(\boldsymbol{x})\mathcal{Q}(\boldsymbol{W'}) + \boldsymbol{b'}) \otimes \mathcal{Q}(\mathcal{Q}(\boldsymbol{x})\mathcal{Q}(\boldsymbol{V'}) + \boldsymbol{c'}) \otimes \sigma'(\mathcal{Q}(\mathcal{Q}(\boldsymbol{x})\mathcal{Q}(\boldsymbol{W'}) + \boldsymbol{b'}))$. After block reconstruction, it is natural to incorporate the smoothing factor $\boldsymbol{s}$ into the weights. The only difference lies in replacing the original activation function $\sigma$ with $\sigma'$, which incurs negligible overhead. Fig 4 presents the output activation distribution of the gated unit in SwiGLU before and after the FSBR. It can be observed that the significant imbalance between channels and tokens, as depicted in Fig 4-a, is effectively alleviated after FSBR (Fig 4-b).

It is worth noting that SmoothQuantXiao et al. (2022) and OmniQuantShao et al. (2023) are subsets of FSBR. FSBR encompasses various previous equivalent quantization algorithms, providing more possibilities for optimizing the distribution of weights and activations. Through mutual optimization across channels, the network demonstrates improved robustness to quantization, as shown in Table 5.

## 3.3 DYNAMIC INTERGER-ONLY MATMUL (DI-MATMUL)

The dynadic arithmetic pipeline is an efficient approach that implements floating-point operations using integer bit-shifts, allowing linear operations to be performed solely through integer arithmetic. Initially developed for convolutional neural networks Jacob et al. (2018), this technique has been extended to Vision Transformers by I-ViT Li & Gu (2023). However, these methods primarily focus on static quantization, where the quantization parameters of the model's activation are fixed during runtime. In the context of LLMs, even after applying inter-channel smoothing, there still exists considerable variation in activation on a token-wise basis, as shown in Fig. 6 in Appendix A. Employing static quantization can result in significant quantization errors and subsequent degradation in accuracy, as depicted in the Fig. 2. Therefore, implementing dynamic quantization for inputs and outputs while adhering to Integer-only constraints poses a significant challenge.

We propose a novel DI-MatMul approach, where the matrix multiplication is formulated as $\boldsymbol{Y^I}, s_y, zp^I = \mathcal{M}(s_1, zp_1^I, \boldsymbol{X_1}, s_2, zp_2^I, \boldsymbol{X_2})$. Herein, $s_1$, $s_2$, and $s_y$ are floating-point scalars representing the quantization steps for the inputs and outputs respectively; while $zp_1^I$ and $zp_2^I$ denote zero-points. To avoid floating-point operations in our method, we represent quantization step $s$ using a dyadic number (DN), i.e., $\frac{m^I}{2^{k^I}}$ where both $m^I$ and $k^I$ are 8-bit integers. Consequently, the entire matrix multiplication can be expressed as:

$$\boldsymbol{Y^I}, m_y^I, k_y^I, zp^I = \mathcal{M}(m_1^I, k_1^I, zp_1^I, \boldsymbol{X_1}, m_2^I, k_2^I, zp_2^I, \boldsymbol{X_2}) \tag{3}$$

For a single matrix multiplication operation:

$$\boldsymbol{P^I} = (\boldsymbol{X_1^I} - zp_1^I)(\boldsymbol{X_2^I} - zp_2^I), \quad \boldsymbol{Y} = \boldsymbol{P^I} \frac{m_1^I m_2^I}{2^{k_1^I + k_2^I}} \tag{4}$$

The intermediate result of the matrix multiplication is denoted as $\boldsymbol{P^I}$. By applying Eq 14 in Appendix and preliminary Eq 3, we obtain the following approximation:

$$\frac{m_y^I}{2^{k_y^I}} \approx s_y = \frac{(p_{\max}^I - p_{\min}^I) \cdot m_1^I \cdot m_2^I}{(2^{n^I} - 1) \cdot 2^{k_1^I + k_2^I}} \tag{5}$$

where $p_{\max}^I$ and $p_{\min}^I$ denote the maximum and minimum values in $\boldsymbol{P^I}$. To obtain the quantization scale of the output:

$$\underset{m_y^I, k_y^I}{\arg\min} \left\| \frac{(p_{\max}^I - p_{\min}^I) \cdot m_1^I \cdot m_2^I}{(2^{n^I} - 1) \cdot 2^{k_1^I + k_2^I}} - \frac{m_y^I}{2^{k_y^I}} \right\|_1 \quad \text{s.t.} \quad m_y^I, k_y^I \in 0, 1, 2, \ldots, 255 \tag{6}$$

Obtaining the optimal values for $m_y^I$ and $k_y^I$ may require multiple iterations. However, by induction, it is known that if an optimal value of $k_0^I$ is achieved for $k_y^I$, then increasing it to $k_0^I + 1$ will not result in a worse outcome. Therefore, we set $m_y^I = 256$ to determine the value of $k_y^I$, and subsequently solve for the remaining variables as follows:

$$k_y^I = \left\lfloor \log_2 \left( \frac{256 \cdot (2^{n^I} - 1) \cdot 2^{k_1^I + k_2^I}}{p_{\max}^I - p_{\min}^I} \right) \right\rfloor = \left\lfloor \log_2 \left( \frac{(2^{n^I} - 1) \cdot 2^{k_1^I + k_2^I + 8}}{p_{\max}^I - p_{\min}^I} \right) \right\rfloor \tag{7}$$

$$m_y^I = \left\lfloor \frac{(p_{\max}^I - p_{\min}^I) \cdot m_{x1}^I \cdot m_{x2}^I}{n^I} \gg (k_1^I + k_2^I - k_y^I) \right\rfloor \tag{8}$$

$$z_y^I = \left\lfloor \frac{-p_{\min}^I \cdot (2^{n^I} - 1)}{p_{\max}^I - p_{\min}^I} \right\rceil, \quad \boldsymbol{Y^I} = \left\lfloor \frac{(\boldsymbol{P^I} - p_{\min}^I) \cdot (2^{n^I} - 1)}{p_{\max}^I - p_{\min}^I} \right\rceil \tag{9}$$

The implementation of $\lfloor \log_2(\cdot) \rfloor$ can be achieved using either the Most Significant Bit (MSB) method or by performing iterative right shifts. As shown in Eq equation 7, equation 8, and equation 9, our approach introduces only a few additional integer-only scalar computations during runtime, making

it more efficient than previous methods. For the Dense layers in Large Language Models (LLMs), one can simply replace an input with its corresponding weights. During inference, since the input usually consists of a single token, this allows for per-token dynamic quantization naturally. In the prefill phase, we adjust the required dimensions accordingly; for instance, we define $\boldsymbol{m^I}$ and $\boldsymbol{k^I}$ as vectors in $\mathbb{R}^{\text{token}}$.

### 3.4 DYNAMIC NON-LINEAR INTEGER-ONLY OPERATIONS

#### 3.4.1 DYNAMIC INTERGER-ONLY CLIPPED SOFTMAX & DYNAMIC INTEGER-ONLY EXPONENT FUNCTION

The Softmax function converts attention scores into a probability distribution. It can be represented as follows:

$$\text{Softmax}\,(x_i) = \frac{e^{x_i}}{\sum_{j=0}^{i} e^{x_j}} = \frac{e^{x_i - x_{\max}}}{\sum_{j=0}^{i} e^{x_j - x_{\max}}}, \quad \text{where } i = 1, 2, \cdots, d \tag{10}$$

We introduce DI-ClippedSoftmax to avoid quantizing the entire input range. As shown in Fig 3, the activation inputs to the Softmax function in LLMs can exhibit significant outliers, with their magnitudes being proportional to the number of tokens. Fortunately, the DI-MatMul approach proposed in Section 3.3 allows for the differences between tokens to be handled individually. However, simply applying 8-bit quantization to the Softmax inputs would result in substantial precision loss, as demonstrated by the ablation studies in Section 6. Starting from the observation that the value of the exponential function at $-20$ can be considered to be zero, this implies that, for each token, only the values within the range $(x_{\max} - 20, x_{\max})$ contribute to the result. For a single token, we can constrain the quantization range by limiting $p_{\min}^I$ as defined in Eq 5. Assuming we want to confine the range within $(p_{\max} - c, p_{\max})$, we first represent $c$ as a dyadic number, i.e., $c = \frac{m_c^I}{2^{k_c^I}}$. Combining this with Eq 5, the entire truncation process can be expressed as follows:

$$p_{min}^I = \max(p_{min}^I, p_{max}^I - c^I) \quad , \text{where } c^I = m_c^I \cdot 2^{k_1^I + k_2^I - k_c^I} \tag{11}$$

Therefore, regardless of the dynamic range, the length of our quantization range will never exceed $c$. In our hyper-parameters tuning experiments (Table 6), we select the optimal value of $c = 15$, ensuring that the maximum quantization error does not exceed 0.03.

We also propose Dynamic Interger-only Exponent function(DI-Exp), an exponential computation method that performs non-linear calculations of the exponential function using only shift operations. For a dynamically quantized input composed of $\boldsymbol{x^I}, m_x^I, k_x^I$, where $\boldsymbol{x^I}$ is already the result after subtracting the maximum value, the computation of the exponential can be expressed as:

$$\mathrm{e}^{\boldsymbol{x^I} \cdot \left(m_x^I \gg k_x^I\right)} = 2^{\boldsymbol{x^I} \cdot \left(m_x^I \gg k_x^I\right) \cdot \log_2 e} = 2^{\boldsymbol{x^I} \cdot m_x^I \cdot \left(\log_2 e \gg k_x^I\right)} \tag{12}$$

We let $s = \frac{\log_2 e}{2^{k_x^I}}$ and $s^I = \left\lfloor \frac{2^{k_x^I}}{\log_2 e} \right\rfloor$, then: $\mathrm{e}^{\boldsymbol{x^I} \cdot \left(m_x^I \gg k_x^I\right)} \approx 2^{\left(\left\lfloor \frac{\boldsymbol{x^I}}{s^I} \right\rfloor + \left(\boldsymbol{x^I} \% s^I\right)s\right)} \approx 2^{-q^I + r^I \cdot s}$

where $q^I = -\left\lfloor \frac{\boldsymbol{x^I}}{s^I} \right\rfloor$, $r^I = \boldsymbol{x^I} \% s^I$ It is known that $r^I \cdot s \in (-1, 0]$, therefore, we can simply perform linear interpolation within this interval. That is, $2^{r^I \cdot s} \approx 1 - \frac{r^I}{2 \cdot s^I}$ Finally, we obtain:

$$\mathrm{e}^{\boldsymbol{x^I} \cdot \left(m_x^I \gg k_x^I\right)} \approx \left(1 - \frac{r^I}{2 \cdot s^I}\right) \gg q^I \tag{13}$$

In DI-Exp, nonlinearity is achieved using only shift operations, which improves computational efficiency compared to complex methods such as quadratic fitting and Taylor series expansion. Only the calculation of $s^I$ and the linear interpolation within a small interval introduce errors, while other computations remain equivalent to the original. Detailed implementation of DI-Exp is in Algorithm1.

#### 3.4.2 DYNAMIC INTERGER-ONLY NORMALIZATION & DYNAMIC INTERGER-ONLY SWIGLU

**DI-Norm**. LayerNorm and RMSNorm are commonly used normalization methods in LLMs. RMSNorm represents a lightweight improvement over LayerNorm, as it does not necessitate the computation of means and biases, thereby reducing computational complexity. Both RMSNorm and LayerNorm exhibit similar characteristics, with significant fluctuations at the channel level. As

depicted in the Fig 3, RMSNorm often exhibits more pronounced differences between channels due to the absence of a centering step. We propose Dynamic Integer-only Normalization (DI-Norm) to adapt to fluctuations between channels. When conducting FSBR, we perform per-channel quantization on the inputs of LayerNorm and RMSNorm. During inference, while computing the mean is straightforward, calculating the variance requires an appropriate root mean square function. Unlike the Newton's method used in I-BERT, to ensure consistency between inference and training, we employ a bit-wise check method as mentioned in the Algorithm 4 to achieve higher precision.

**DI-SwiGLU**. During the block reconstruction stage, we decompose the SiLU activation function of SwiGLU into $\text{SiLU}(\boldsymbol{x}) = \boldsymbol{x} \cdot \sigma(\boldsymbol{x})$ to achieve non-linear smoothing. As described in Algorithm 3, we implement the Dynamic Integer-only SwiGLU (DI-SwiGLU), consisting of a single sigmoid nonlinearity and two multiplication operations, where the sigmoid implementation involves multiple invocations of our proposed DI-Exp operator.

## 4 EXPERIMENTS

**Implementation Details.** I-LLM ensures that the activation of all non-linear operators remains at 8 bits, while the weights and activations of linear operators are determined by the current quantization configuration, such as W4A8. Consistent with other methods, we employ 128 samples for reconstruction. During the reconstruction phase, we maintain that the input to Softmax is not quantized and ensure that all smoothing coefficients maintain a common learning rate of $5 \times 10^{-3}$. After the reconstruction, all operators will be replaced with respective versions supporting dynamic integer-only inference. All experiments are conducted on Nvidia A6000 GPU.

**Models & Evaluation Metric.** We conduct experiments on several commonly used open-source LLMs, including OPT (Zhang et al., 2022), LLaMA (Touvron et al., 2023a), and LLaMA2 (Touvron et al., 2023b). Additionally, we also evaluated the recently impressive LLaMA3-8B model. For the sake of comparison, we tested the impact of quantization on Perperxity on two of the most commonly used datasets WikiText2 (Merity et al., 2016) and C4 (Raffel et al., 2020). Moreover, accuracy is evaluated in zero-short tasks including PIQA (Bisk et al., 2020), ARC (Boratko et al., 2018), BoolQ (Clark et al., 2019), HellaSwag (Zellers et al., 2019), Winogrande (Sakaguchi et al., 2021).

### 4.1 QUANTITATIVE RESULTS

Notably, our work is the first to address integer-only quantization for LLMs, whereas all the methods we compare against in our experiments are not integer-only except I-Bert. Although this may seem somewhat unfair for us, I-LLM still demonstrates the superiority through its remarkable performance.

Fig 2 illustrates the efficacy of our 8-bit quantization technique on widely adopted large-scale language models. Despite SmoothQuant's prior success with 8-bit quantization, our approach demonstrates performance for each model that is closer to floating-point precision. This suggests that even with low-bit quantization using integer-only conditions (e.g., 8-bit), we can achieve performance comparable to floating-point representation. These findings strongly validate the effectiveness of our proposed I-LLM, as it enables integer-only operators to yield results highly akin to those obtained through floating-point operations, using simple arithmetic and integer bit-shifting.

**Quantization results on OPT and LLaMA.** As shown in Table 1 and Table 2, we report the perplexity performance of I-LLM on the C4 and WikiText2 datasets. As depicted in these tables, I-LLM consistently surpasses previous methods across a diverse array of LLM families (OPT 6.7B-30B, LLaMA-1 7B-30B, LLaMA-2 7B-13B, LLaMA-3 8B) and varying levels of precision. Particularly noteworthy is its performance under the W4A4 setting, where our proposed method achieves perplexity values that are consistently 10% to 30% lower than those attained by state-of-the-art methods.

Table 1: Quantitative weight-activation quantization PPL($\downarrow$) results on OPT Family of I-LLM.

| #Bits | Method | OPT-6.7B | | OPT-13B | | OPT-30B | |
|---|---|---|---|---|---|---|---|
| | | WikiText2 | C4 | WikiText2 | C4 | WikiText2 | C4 |
| FP16 | - | 10.86 | 11.74 | 10.13 | 11.20 | 9.56 | 10.69 |
| W6A6 | SmoothQuant | 11.34 | 12.14 | 10.56 | 11.40 | 9.67 | 10.81 |
| | RPTQ | 11.19 | 12.08 | 11.00 | 11.68 | 10.22 | 11.73 |
| | OmniQuant | 10.96 | 11.81 | 10.21 | 11.27 | 9.62 | 10.76 |
| | **I-LLM** | 10.94 | 11.82 | 10.17 | 11.90 | 9.72 | 10.83 |
| W4A4 | SmoothQuant | 1.8e4 | 1.5e4 | 7.4e3 | 5.6e3 | 1.2e4 | 8.3e3 |
| | RPTQ | 12.00 | 12.85 | 12.74 | 14.71 | 11.15 | 13.48 |
| | OmniQuant | 12.24 | 13.56 | 11.65 | 13.46 | 10.60 | 11.90 |
| | **I-LLM** | 12.20 | 12.21 | 11.45 | 13.41 | 10.53 | 11.66 |

Table 2: Quantitative weight-activation quantization PPL(↓) results of I-LLM. We report WikiText2 and C4 perplexity of LLaMA Family in this table.

| #Bits | Method | LLaMA-7B | | LLaMA-13B | | LLaMA-30B | | LLaMA2-7B | | LLaMA2-13B | | LLaMA3-8b | |
|---|---|---|---|---|---|---|---|---|---|---|---|---|---|
| | | WikiText2 | C4 | WikiText2 | C4 | WikiText2 | C4 | WikiText2 | C4 | WikiText2 | C4 | WikiText2 | C4 |
| FP16 | - | 5.68 | 7.08 | 5.09 | 6.61 | 4.10 | 5.98 | 5.47 | 6.97 | 4.88 | 6.46 | 6.14 | 8.88 |
| W6A6 | SmoothQuant | 6.03 | 7.47 | 5.42 | 6.97 | 4.55 | 6.34 | 6.2 | 7.76 | 5.18 | 6.76 | 7.08 | 10.16 |
| | OmniQuant | 5.96 | 7.43 | 5.28 | 6.84 | 4.38 | 6.22 | 5.87 | 7.48 | 5.14 | 6.74 | 6.97 | 10.08 |
| | **I-LLM** | 5.84 | 7.32 | 5.23 | 6.79 | 4.32 | 6.25 | 5.68 | 7.27 | 5.10 | 6.74 | 6.61 | 9.77 |
| W4A4 | SmoothQuant | 22.25 | 32.32 | 40.05 | 47.18 | 192.40 | 122.38 | 83.12 | 77.27 | 35.88 | 43.19 | 418.88 | 312.86 |
| | OmniQuant | 11.26 | 14.51 | 10.87 | 13.78 | 10.33 | 12.49 | 14.26 | 18.02 | 12.30 | 14.55 | 437.88 | 315.69 |
| | AffineQuant | 10.28 | 13.64 | 10.32 | 13.44 | 9.35 | 11.58 | 12.69 | 15.76 | 11.45 | 13.97 | - | - |
| | **I-LLM** | 9.10 | 12.33 | 7.99 | 10.96 | 7.24 | 9.85 | 10.44 | 12.92 | 9.76 | 12.57 | 21.19 | 30.9 |

Table 3: The performance of various methods for 4-bit and 6-bit quantization on the LLaMA family models across six zero-shot datasets.

| LLaMA / Acc(↑) | #Bits | Method | PIQA(↑) | ARC-e(↑) | ARC-c(↑) | BoolQ(↑) | HellaSwag(↑) | Winogrande(↑) | Avg.(↑) |
|---|---|---|---|---|---|---|---|---|---|
| LLaMA-7B | FP16 | - | 77.47 | 52.48 | 41.46 | 73.08 | 73.00 | 67.07 | 64.09 |
| | W6A6 | SmoothQuant | 76.75 | 51.64 | 39.88 | 71.75 | 71.67 | 65.03 | 62.81 |
| | W6A6 | OmniQuant | 77.09 | 51.89 | 40.87 | 72.53 | 71.61 | 65.03 | 63.17 |
| | W6A6 | **I-LLM** | 76.99 | 52.66 | 40.78 | 72.94 | 71.31 | 65.67 | **63.39** |
| | W4A4 | SmoothQuant | 49.80 | 30.40 | 25.80 | 49.10 | 27.40 | 48.00 | 38.41 |
| | W4A4 | LLM-QAT | 51.50 | 27.90 | 23.90 | 61.30 | 31.10 | 51.90 | 41.27 |
| | W4A4 | LLM-QAT+SQ | 55.90 | 35.50 | 26.40 | 62.40 | 47.80 | 50.60 | 46.43 |
| | W4A4 | OS+ | 62.73 | 39.98 | 30.29 | 60.21 | 44.39 | 52.96 | 48.43 |
| | W4A4 | OmniQuant | 66.15 | 45.20 | 31.14 | 63.51 | 56.44 | 53.43 | 52.65 |
| | W4A4 | AffineQuant | 69.37 | 42.55 | 31.91 | 63.73 | 57.65 | 55.33 | 53.42 |
| | W4A4 | **I-LLM** | 67.25 | 45.58 | 32.59 | 63.88 | 58.89 | 57.06 | **54.21** |
| LLaMA-13B | FP16 | - | 79.10 | 59.89 | 44.45 | 68.01 | 76.21 | 70.31 | 66.33 |
| | W6A6 | SmoothQuant | 77.91 | 56.60 | 42.40 | 64.95 | 75.36 | 69.36 | 64.43 |
| | W6A6 | OmniQuant | 78.40 | 57.28 | 42.91 | 67.00 | 75.82 | 68.27 | 64.95 |
| | W6A6 | **I-LLM** | 77.48 | 56.94 | 44.03 | 64.92 | 75.24 | 69.14 | **64.63** |
| | W4A4 | SmoothQuant | 61.04 | 39.18 | 30.80 | 61.80 | 52.29 | 51.06 | 49.36 |
| | W4A4 | OS+ | 63.00 | 40.32 | 30.38 | 60.34 | 53.61 | 51.54 | 49.86 |
| | W4A4 | OmniQuant | 69.69 | 47.39 | 33.10 | 62.84 | 58.96 | 55.80 | 54.37 |
| | W4A4 | AffineQuant | 66.32 | 43.90 | 29.61 | 64.10 | 56.88 | 54.70 | 52.58 |
| | W4A4 | **I-LLM** | 67.95 | 48.15 | 34.47 | 62.29 | 63.13 | 59.98 | **56.00** |
| LLaMA-30B | FP16 | - | 80.08 | 58.92 | 45.47 | 68.44 | 79.21 | 72.53 | 67.44 |
| | W6A6 | SmoothQuant | 77.14 | 57.61 | 42.91 | 65.56 | 78.07 | 69.92 | 65.20 |
| | W6A6 | OmniQuant | 78.40 | 57.28 | 42.91 | 67.00 | 75.82 | 68.27 | 64.95 |
| | W6A6 | **I-LLM** | 79.43 | 58.88 | 45.14 | 73.36 | 78.51 | 72.61 | **67.99** |
| | W4A4 | SmoothQuant | 58.65 | 35.53 | 27.73 | 60.42 | 35.56 | 48.06 | 44.83 |
| | W4A4 | OS+ | 67.63 | 46.17 | 34.40 | 60.70 | 54.32 | 52.64 | 52.62 |
| | W4A4 | OmniQuant | 71.21 | 49.45 | 34.47 | 65.33 | 64.65 | 59.19 | 56.63 |
| | W4A4 | AffineQuant | 66.32 | 43.90 | 29.61 | 64.10 | 56.88 | 54.70 | 52.58 |
| | W4A4 | **I-LLM** | 71.38 | 51.81 | 37.12 | 65.69 | 67.79 | 61.40 | **59.20** |

**Zero-shot results on LLaMA.** Table 3 presents our performance on six zero-shot tasks, employing both W4A4 and W6A6 settings. Particularly striking is the performance of the LLaMA-30b model, configured with full quantization at W6A6 precision, which achieves an average accuracy on these tasks surpassing even that of the floating-point model. This achievement underscores the potential of fully quantized integer-only methods in maintaining the generalization capabilities of LLMs to a considerable degree.

Table 4: Performance comparison of LLaMA models using I-LLM and traditional methods. The table presents the comparison between the full-precision (Fp16) and quantized (W4A4) methods, including the proposed I-LLM W4A4 approach. I-LLM W4A4 achieves up to a **2.63x** speedup and a **3.08x** memory saving across different LLaMA model sizes (7B, 13B, 8B). All experiments used a token length of 2048.

| Model | Method | Running Memory | Latency(ms) ↓ | SpeedUp | Memory Saving |
|---|---|---|---|---|---|
| LlaMA2-7b | Fp16 | 14.40G | 649 | - | - |
| | Traditional W4A4 | 5.36G | 314 | 2.06x | 2.69x |
| | **I-LLM W4A4** | 4.73G | 260 | 2.49x | 3.04x |
| LlaMA2-13b | Fp16 | 27.03G | 1240 | - | - |
| | W4A4 | 9.31G | 592 | 2.09x | 2.90x |
| | **I-LLM W4A4** | 8.76G | 493 | 2.52x | 3.08x |
| LlaMA3-8b | Fp16 | 16.22G | 702 | - | - |
| | W4A4 | 7.43G | 316 | 2.22x | 2.18 |
| | **I-LLM W4A4** | 3.74G | 267 | 2.63x | 3.03x |

**Latency and memory saving on LLaMA.** To validate the versatility of I-LLM, we have executed a series of tests to assess its performance enhancement on Nvidia A6000 GPU. In Table 4, we analyze the latency and memory saving of I-LLM across various models. Under W4A4 settings, I-LLM achieves a substantial 2.6x speed boost and a 3x memory saving compared to the fp16 model, surpassesing other W4A4 methods in both speed and memory saving. Specifically, as shown in Table 7, DI-MatMul outperforms conventional quantized GEMM in both weight-activation (1.2x speedup) and activation-activation (2.1x speedup) scenarios, attributed to DI-MatMul's efficient integer arithmetic and its requant-only feature. Additionally, Table 8 shows that DI-ClippedSoftmax is over 5x faster than the fp16 implementation across different input sizes, while DI-Norm also exhibits lower latency. Overall, I-LLM significantly improves both speed and memory saving, outperforming competing methods and demonstrating its superior performance.

## 4.2 Ablation Study

**Contribution of Fully-Smooth Block Reconstruction** In Table 5, we meticulously assess the impact of various PTQ methods on model accuracy. To maintain impartiality, this experiment abstains from utilizing integer-only operators; instead, all quantization procedures are substituted with pseudo-quantization. The nodes necessitating quantization align with those delineated in Fig 1. Notably, conventional non-integer-only methodologies often overlook the influence of activation in non-linear layers, leading to compromised model accuracy during integer inference. However, with the integration of FSBR, these activations are thoughtfully considered during PTQ, effectively reinstating the model's accuracy under full quantization.

Table 5: Impact of different PTQ methods and integer-only operators on LLaMA-7B PPL($\downarrow$) across WikiText2 and C4 datasets.

| LLaMA-7B | W4A4 | | W6A6 | |
|---|---|---|---|---|
| Method | WikiText2 | C4 | WikiText2 | C4 |
| SmoothQuant | 256.58 | 218.47 | 6.09 | 7.6 |
| OmniQuant | 122.18 | 183.2 | 5.99 | 7.57 |
| **FSBR** | **9.44** | **12.72** | **5.83** | **7.02** |
| +DI-CLippedSoftamx | 9.44 | 12.72 | 5.83 | 7.02 |
| +DI-Swiglu | 9.12 | 12.38 | 5.83 | 7.04 |
| +DI-Norm | 9.52 | 12.63 | 5.85 | 7.35 |

Table 6: Effect of clipped value in DI-ClippedSoftmax.

| LLaMA-7B | W4A4 | | W6A6 | |
|---|---|---|---|---|
| Clipped Value $c$ | WikiText2 | C4 | WikiText2 | C4 |
| – | 7360945.00 | 1998371.38 | 60335.22 | 47103.44 |
| 20 | 9.15 | 12.39 | 5.86 | 7.36 |
| 17 | 9.19 | 12.38 | 5.86 | 7.37 |
| **15** | **9.16** | **12.36** | **5.85** | **7.36** |
| 12 | 9.19 | 12.35 | 5.86 | 7.36 |
| 10 | 9.23 | 12.45 | 5.89 | 7.42 |

**Impact of Integer-Only Operators.** As shown in Table 5, we intricately outline the precise impact of each integer-only operator on the holistic accuracy of the model. Additionally, we ablate the influence of the clipping coefficient $c$ in Eq 11 within DI-ClippedSoftmax on model accuracy, as depicted in Table 6. Notably, owing to residual connections, the quantization of DI-Norm engenders a discernible decrement in performance, a phenomenon meticulously anticipated in our comprehensive analysis. In Table 5, we elaborate on the influence of each integer-only operator on the overall model accuracy. Specifically, Table 6 illustrates the impact of the clipping coefficient $c$ in the Eq 11 from DI-ClippedSoftmax on model accuracy. It is worth noting that the quantization of DI-Norm leads to a performance decline, primarily due to outliers in the residual connections.

## 5 Conclusion

In this paper, we present I-LLM, a fully-quantized integer-only PTQ framework for LLMs. We address the challenge of fluctuating activations in both linear and non-linear operations by proposing Fully-Smooth Block-Reconstruction (FSBR) to harmonize inter-channel variations and Dynamic Integer-only MatMul (DI-MatMul) to handle inter-token variations. Additionally, we design DI-ClippedSoftmax, DI-Exp, and DI-Norm as lightweight integer-only operators to replace computationally intensive floating-point operations. Experiments demonstrate that I-LLM outperforms simulated quantization methods and achieves comparable accuracy to the floating-point baseline. Notably, I-LLM operates effectively in a low-bit quantization setting with minimal accuracy loss, thereby bridging the gap between low-precision integer-only quantization and full-precision LLMs. Moreover, experiments demonstrate that I-LLM not only outperforms simulated quantization methods but also achieves comparable accuracy to the floating-point baseline. In practical scenarios, I-LLM benefits from the performance gains of low-precision computation, while its consistent operation at low bit-width significantly reduces memory utilization. Consequently, this work not only accelerates LLM inference on server-side deployments but also paves the way for efficient LLM deployment on edge devices without floating-point capabilities.

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

## A    APPENDIX

### A.1    QUANTIZATION PRELIMINARIES

**Quantization & Dequantization.** Quantization typically refers to mapping a floating-point number to a discrete interval with integer number. Here, we only consider uniform quantization. The quantization process can be expressed as follows:

$$\boldsymbol{X}^I = \text{clamp}\left(\left\lfloor \frac{\boldsymbol{X}}{s} \right\rceil + zp^I, 0, 2^{n^I} - 1\right) \tag{14}$$

$$s = \frac{x_{\max} - x_{\min}}{2^{n^I} - 1} \tag{15}$$

$$zp^I = \left\lfloor \frac{-x_{\min}}{s} \right\rceil \tag{16}$$

$$\boldsymbol{X}' = (\boldsymbol{X}^I - zp^I) \cdot s \tag{17}$$

where $\boldsymbol{X}$ is the floating-point tensor, $\boldsymbol{X}^I$ is its quantized counterpart and $\boldsymbol{X}'$ is the dequantized resuls of $\boldsymbol{X}^I$. Here, $n^I$ represents the number of bits (e.g., 8). $s$ is the quantization step size, determined by $x_{\min}, x_{\max}$, and $n^I$.clamp represents truncation function. The choice of $s$ greatly affects the accuracy of the quantized model. We can obtain $s$ from the activations of some samples, which is called static quantization. Alternatively, we can derive it from runtime statistics, known as dynamic quantization. Quantization can also be distinguished by its granularity into per-channel quantization and per-token quantization (Yao et al., 2022).

### A.2    IMPLEMENTATION DETAILS OF FULLY-SMOOTH BLOCK RECONSTRUCTION

To be more specific, the core idea of FSBR is to reduce quantization errors and enhance the overall performance of the model by smoothing the distribution of activations and weights across different computational units, such as linear operations and nonlinear activation functions. In contrast to existing methods, such as SmoothQuant and OmniQuant, FSBR focuses on all computational units and smooths all suitable activation-activation and activation-weight pairs.

Additionally, FSBR introduces unique contributions that are not present in existing state-of-the-art (SOTA) methods, including:

- **Linear Smoothing**:
  - **GQA**: FSBR smooths the Value Projection and Output Projection layers, as shown in the third method in Figure 5. Specifically, head-wise smoothing is applied between the Value Projection and Output Projection layers. For each head, the interaction between the Value Projection and Output Projection layers is given by

    $$\boldsymbol{P}\boldsymbol{x}_v^h \boldsymbol{W}_v^h \boldsymbol{W}_o^h$$

    which can be rewritten as

    $$\boldsymbol{P}\boldsymbol{x}_v^h \boldsymbol{W}_v^h \boldsymbol{S}_{ov}^h {\boldsymbol{S}_{ov}^{-1}}^h \boldsymbol{W}_o^h$$

    Here, $\boldsymbol{P}$ represents the attention matrix, and $\boldsymbol{S}_{ov}$ is the learnable scaling matrix. This approach simultaneously smooths the channel-wise differences between the output activations of the Value Projection and the input activations of the Output Projection.
  - **Up and Down Projection Smoothing**: As shown in the third method in Figure 5, FSBR also learns smoothing coefficients between the Up and Down projections. This smoothing is based on the invariance property

    $$\boldsymbol{x}_1 \boldsymbol{W}_1 \otimes \boldsymbol{x}_2 \boldsymbol{W}_2 = \boldsymbol{x}_1 \boldsymbol{W}_1 \boldsymbol{S} \otimes \boldsymbol{x}_2 \boldsymbol{W}_2 \boldsymbol{S}^{-1}$$

    This ensures smoothness in the channel-wise differences between the output activations of the Up Projection and the input activations of the Down Projection.
- **Nonlinear Activation Smoothing**:

- **SiLU Activation Smoothing**: FSBR introduces the NonLinear Act-Smooth method, which smooths the channel-wise differences between two inputs, as depicted in the fourth method in Figure 5. Specifically, we introduce a smoothing parameter $s$ to reconstruct the two inputs $x_1$ and $x_2$ into a form that is more suitable for quantization (see Equation 2). This innovation significantly reduces quantization errors in the SwiGLU activation values (see Figure 4).

- **Softmax Input Activation Smoothing**: In Softmax, the outlier problem in activation values is particularly noticeable. FSBR mitigates this by dynamically truncating the input range, keeping the Softmax input within $(-c, 0)$ (see Equation 11). This adjustment limits the quantization range and reduces the quantization errors in the Softmax input.

## A.3 DYNAMIC INTERGER-ONLY ALGORITHMS

---

**Algorithm 1** Dynamic Integer-only Exp: DI-Exp

---

**Input**: Integer input $\boldsymbol{x}^I$, Integer input scale factor $m_x^I$ and Integer shift factor $k_x^I$.
**Constant**: Integer $m_e^I$ and $k_e^I$ that satisfy $m_e^I >> k_e^I = log_2 e$. Integer $b_y^I$ represents the bitwidth of output, e.g., $b_y^I = 16$
**Output**: The Integer result $\boldsymbol{y}^I$ of the exponential function amplified by $2^{b^I}$

1: $n^I = k_e^I + k_x^I$
2: $\boldsymbol{x}^I = -(\boldsymbol{x}^I \cdot m_x^I \cdot m_e^I)$
3: $\boldsymbol{p}^I = \boldsymbol{x}^I \gg n^I$
4: $\boldsymbol{r}^I = \boldsymbol{x}^I - \boldsymbol{p}^I \ll n^I$
5: $\boldsymbol{y}^I = (1 \ll n^I) - (\boldsymbol{r}^I \gg 1)$
6: $\boldsymbol{y}^I = \boldsymbol{y}^I \gg (\boldsymbol{p}^I - b_y^I)$
7: **return** $\boldsymbol{y}^I$

---

**Algorithm 2** Dynamic Integer-only Softmax : DI-Softmax

---

**Input**: Integer input $\mathbf{x}^I$, Integer input scale factor $m_x^I$, Integer shift factor $k_x^I$.
**Output**: Integer output $\mathbf{y}^I$, Integer output scale factor $m_y^I$ and Integer shift factor $k_y^I$.

1: $\mathbf{x}_\Delta^I = \mathbf{x}^I - \max(\mathbf{x}^I)$
2: $\mathbf{e}_\Delta^I = \text{DI-Exp}(\mathbf{x}_\Delta^I, m_x^I, k_x^I)$
3: $e_{sum}^I = \sum \boldsymbol{e}_\Delta^I$
4: $\boldsymbol{o}^I = \lfloor \dfrac{\boldsymbol{e}_\Delta^I \ll b_o^I}{e_{sum}^I} \rfloor$
5: **return** $\boldsymbol{o}^I, 1, k_y^I$

---

**Algorithm 3** Dynamic Integer-only SwiGLU : DI-SwiGLU

---

**Input**: Integer input $\mathbf{x}_{gate}^I, \mathbf{x}_{up}^I$, Integer input scale factor $m_{gate}^I, m_{up}^I$, Integer shift factor $k_{gate}^I, k_{out}^I$, smooth factor $\alpha_{smooth}$ and output bit-precision $p_{out}^I$.
**Output**: Integer output $\mathbf{y}_{out}^I$, Integer output scale factor $m_{out}^I$ and Integer shift factor $k_{out}^I$.

1: $\mathbf{x}_{smoothed\_gate}^I = \mathbf{x}_{gate}^I / \alpha_{smooth}$
2: $\mathbf{x}_\Delta^I = \mathbf{x}_{smooth\_gate}^I - max(\mathbf{x}_{smoothed\_gate}^I)$
3: $\mathbf{Exp}_\Delta^I = \text{DI-Exp}(\mathbf{x}_\Delta^I, m_{gate}^I, k_{gate}^I)$
4: $\mathbf{Exp}_{-max}^I = \text{DI-Exp}(-max(\mathbf{x}_{smoothed\_gate}^I), m_{gate}^I, k_{gate}^I)$
5: $\mathbf{y}_{out}^I = \mathbf{x}_{gate}^I \cdot IntDiv(\mathbf{Exp}_\Delta^I, \mathbf{Exp}_\Delta^I + \mathbf{Exp}_{-max}^I, p_{out}^I) \cdot \mathbf{x}_{up}^I$
6: $m_{out}^I = m_{gate}^I \cdot m_{up}^I$
7: $k_{out}^I = k_{gate}^I + k_{up}^I + p_{out}^I - 1$
8: **return** $\mathbf{y}_{out}^I, m_{out}^I, k_{out}^I$

---

---

**Algorithm 4** Dynamic Integer-only RMSnorm : DI-RMSnorm

---

**Input**: Integer input $\mathbf{x}_{in}^I$, input per-channel Integer scale factor $\mathbf{m}_{in}^I$, input per-channel Integer shift factor $\mathbf{k}_{in}$, weights of RMSnorm $\boldsymbol{\gamma}$ and output bit-precision $p_{out}^I$.
**Output**: Integer output $\mathbf{y}_{out}^I$, and Output scale factor $\mathbf{S}_{out}$.

1: **function** I-SQRT($I_{in}$)
2:     $v = 15$
3:     $n = 1$
4:     $b = 0\,x\,8000$
5:     **while** b **do**
6:         $temp = ((n << 1) + b) << v - -$
7:         **if** $I_{in} \geq temp$ **then**
8:             $n+ = b$
9:             $I_{in}- = temp$
10:        **end if**
11:        $b >>= 1$
12:    **end while**
13:    $I_{out} = n$
14:    **return** $I_{out}$
15: **end function**
16:
17: **function** DI-RMSNORM($\mathbf{x}_{in}^I, \mathbf{m}_{in}^I, \mathbf{k}_{in}^I, \boldsymbol{\gamma}, p_{out}^I$)
18:    $\mathbf{shift}^I = \mathbf{k}_{in}^I - min(\mathbf{k}_{in}^I)$
19:    $\mathbf{m}^I = \mathbf{m}_{in}^I \ll \mathbf{shift}^I$
20:    $m_{min} = min(m)$
21:    $s, n = shape(\mathbf{x}_{in}^I)$
22:    $\boldsymbol{\alpha}^I = round(\mathbf{m}^I \cdot M / m_{min})$
23:    $var^I = (sum(\mathbf{x}_{in}^I * \mathbf{m}^I))^2$
24:    $std^I = \text{I-SQRT}(var^I)$
25:    $dim\_sqrt^I = \text{I-SQRT}(n)$
26:    $\mathbf{y}_{out}^I = IntDiv(\mathbf{x}_{in}^I \cdot N \cdot dim\_sqrt^I, std^I, p_{out}^I)$
27:    $\mathbf{S}_{out} = \mathbf{m}^I \cdot \boldsymbol{\alpha}^I \cdot \boldsymbol{\gamma} / (m_{min} \cdot N)$
28:    **return** $\mathbf{y}_{out}^I, \mathbf{S}_{out}$
29: **end function**

---

## A.4 THE OVERHEAD AND EFFICIENCY OF DI-MATMUL

The DI-MatMul has lower overhead and higher execution efficiency compared to conventional GEMM quantization. As detailed in Section 3.3 of paper, we utilize two 8-bit scalars, $m^I$ and $k^I$ to represent the quantization scale for a batch of activations. This approach incurs no additional storage overhead compared to the previous method using FP16 to store the quantization scale. After obtaining the intermediate results from integer-only matrix operations, I-LLM only requires a small amount of scalar integer operations and one vector-scalar multiplication, as specified in Formulas 6, 7, and 8. The output of DI-Matmul is composed of a low-bit-width integer matrix and three 8-bit scalars.

It should be noted that conventional quantization methods still require a large number of floating-point computations during inference. In contrast, all computations in DI-MatMul are integer-based, effectively leveraging efficient integer arithmetic units. To be more specific:

- Weight-only quantization demands an online dequantization step for the quantized weights, involving floating-point multiplication as a precursor to the subsequent floating-point General Matrix Multiply (GEMM) operations.

- The joint quantization of weights and activations often incurs the necessity for online quantization and dequantization procedures specifically for the activations. More importantly, the output of traditional quantization methods is typically rendered in floating-point format (e.g., FP16), while the output of DI-MatMul is in a low-bit-width integer format. This paradigm not only conserves considerable memory and bandwidth but also reduces latency. For instance, compared to FP16 format, employing 4-bit quantization for feature-map and KV-cache results in a substantial reduction of memory and bandwidth usage by 75

- As shown in Table 7, DI-MatMul outperforms conventional quantized GEMM in both weight-activation (1.2x speedup) and activation-activation (2.1x speedup) scenarios, attributed to DI-MatMul's efficient integer arithmetic and its requant-only feature.

Table 7: The latency comparison of Quantized-Gemm(QGemm) and DI-MatMul in both weight-activation and activation-activation scenarios. Both QGemm and DI-MatMul are implemented based on NVIDIA Cutlass.

|  | Input Size | QGemm(4bit) | DI-MatMul(4bit) | Speedup |
|---|---|---|---|---|
| Act-Act | 1 x 4096 | 1.14ms | 0.45ms | 2.53x |
|  | 2048 x 4096 | 1.71ms | 0.69ms | 2.48x |
|  | 1 x 8192 | 3.73ms | 1.54ms | 2.42x |
|  | 4096 x 8192 | 5.92ms | 2.79ms | 2.12x |
|  | 8192 x 8192 | 8.21ms | 3.83ms | 2.14x |
| Act-Weight | 1 x 8192 | 0.29ms | 0.21ms | 1.38x |
|  | 4096 x 8192 | 2.08ms | 1.75ms | 1.18x |
|  | 8192 x 8192 | 3.91ms | 3.07ms | 1.27x |

## A.5 FULL RESULTS

### A.5.1 I-LLM IN DIFFERENT QUANTIZATION CONFIGURATIONS

I-LLM is a framework designed for fully integer quantization of large language models. However, FSBR demonstrates significant performance improvements across various quantization configurations. In Table 9, we present the results for different settings, including W4A16 and W4A8. The experimental results clearly demonstrate the effectiveness of FSBR in restoring the quantization accuracy of the model.

### A.5.2 MORE COMPARISONS WITH OTHER METHODS

It is important to note that conventional quantization methods (Xiao et al., 2022; Shao et al., 2023),still rely on a significant amount of floating-point computations during inference. Furthermore, while

Table 8: Latency comparison of different Softmax or Norm implementation methods. FP16 represents the native implementation in Torch. I-LLM achieves a twofold speedup for Softmax and accelerates Norm, benefiting from efficient integer computation and reduced memory transfer overhead.

| | Input Size | FP Latency | I-LLM Latency | SpeedUp |
|---|---|---|---|---|
| Softmax | 1 x 4096 | 1.89ms | 0.28ms | 6.75x |
| | 2048 x 4096 | 1.91ms | 0.33ms | 5.79x |
| | 4096 x 4096 | 1.99ms | 0.41ms | 4.85x |
| | 131072 x 4096 | 10.75ms | 5.90ms | 1.82x |
| RMSNorm | 1 x 4096 | 0.35ms | 0.35ms | 1.01x |
| | 2048 x 4096 | 0.23ms | 0.19ms | 1.21x |
| | 4096 x 4096 | 0.36ms | 0.23ms | 1.54x |
| | 1 x 8192 | 0.08ms | 0.07ms | 1.08x |
| | 4096 x 8192 | 0.67ms | 0.40ms | 1.68x |
| | 8192 x 8192 | 3.18ms | 2.97ms | 1.07x |

Table 9: Accuracy performance of the Llama2 model under different quantization configurations using the I-LLM method.

| Quantization Setting | Wikitext2 PPL | C4 PPL |
|---|---|---|
| W16A16 | 5.47 | 6.97 |
| W4A16 | 5.66 | 7.24 |
| W6A6 | 5.68 | 7.27 |
| W4A8 | 5.69 | 7.28 |
| W4A4 | 9.60 | 12.71 |
| W3A8 | 11.29 | 18.02 |
| W2A8 | 25.33 | 35.07 |

some existing methods have achieved INT4 precision with minimal accuracy loss, such as AWQ (Lin et al., 2023), these approaches primarily focus on weight-only quantization. A crucial distinction is that in weight-only quantization methods, both the input and output activations are typically represented in floating-point format (e.g., FP16). In contrast, I-LLM applies low-bit quantization to the inputs and outputs of all operators throughout the entire dataflow, including GEMM (e.g., FC, Matmul), Norm, Softmax, SiLU, and Add. Therefore, direct comparisons between I-LLM and weight-only quantization methods are not entirely fair, as I-LLM faces significantly greater challenges in the quantization process. Despite this, I-LLM has achieved notable, and in some cases, even superior results compared to these methods, as demonstrated in Tab32 10. In addition to traditional quantization, we also report comparative results with the low-bit floating-point method, LLM-FP4 (Liu et al., 2023b), as shown in Table 11.

Table 10: Comparison of perplexity between I-LLM and AWQ (Lin et al., 2023) methods on WikiText2. Note that AWQ is a weight-only quantizationmethod that only quantizes the weights of linear layers, while I-LLM considers quantization of all weights and activations in the network.

| QuantSetting | Method | LLAMA3 8B | LLAMA2 7B | LLAMA2 13B |
|---|---|---|---|---|
| - | FP16 | 6.14 | 5.47 | 4.95 |
| W4 only | AWQ | 7.04 | 6.02 | 5.06 |
| W6 only | AWQ | 6.21 | 5.49 | 4.89 |
| W6A6 | AWQ | 6.90 | 5.98 | 5.19 |
| | **I-LLM** | 6.61 | 5.68 | 5.10 |
| W4A4 | AWQ | 125.74 | 8248 | 133.62 |
| | **I-LLM** | 21.19 | 10.44 | 9.76 |

Table 11: Comparison of I-LLM with current state-of-the-art low-bit floating-point quantization methods on six zero-shot tasks.Note that LLM-FP4 only quantizes the weights and activations of linear layer.

|  |  | Quant Setting | PIQA | ARC-e | Arc-c | BoolQ | HellaSwag | Winogrande | Avg |
|---|---|---|---|---|---|---|---|---|---|
| llama-7b | FP16 | - | 77.47 | 52.48 | 41.46 | 73.08 | 73.00 | 67.07 | 64.10 |
|  | LLM-FP4 | W4A4 | 71.40 | 63.80 | 31.50 | 62.10 | 45.70 | 62.60 | 56.00 |
|  | **I-LLM** | W4A4 | 67.00 | 47.35 | 32.94 | 63.88 | 58.49 | 58.80 | 54.74 |
| llama-13b | FP16 | - | 79.10 | 59.89 | 44.45 | 68.01 | 76.21 | 70.31 | 66.33 |
|  | LLM-FP4 | W4A4 | 72.20 | 69.10 | 37.30 | 69.30 | 50.70 | 64.10 | 60.60 |
|  | **I-LLM** | W4A4 | 67.95 | 48.15 | 34.47 | 62.29 | 63.13 | 59.98 | 56.00 |

## A.6  LIMITATION AND DISCUSSION

We have shown evidence that our I-LLM can replicate float-point performance with an integer-only module with 8-bit or lower bits. A limitation is that we only focused on natural language models, however, it would be interesting to explore how I-LLM performs in computer vision tasks. We leave this for future work.

## A.7 VISIBLE RESULTS

In this section, we provide a comprehensive presentation of our results across various datasets to complement the main paper. Specifically, the results include:

1. Figure 6 illustrates the activation distributions of the QKV Projection and Up Projection in LLaMA-2-7B. It is evident that the disparities between activation channels have been mitigated after training with FSBR.

2. Large language models exhibit significant outliers at the token level. Figure 7 illustrates token-level outliers in the input down_proj of the first layer of the LLaMA-2-7B and LLaMA-3-8B models. These outliers arise from various factors, with special positions often triggering their occurrence. For instance, as shown in Figure 7, the first token (token 0) frequently exhibits anomalous behavior. Additionally, as shown in Figure 8 certain special tokens or attention distributions can lead to severe token-level outliers, where extreme outliers for a small subset of tokens far exceed the variability across channels.

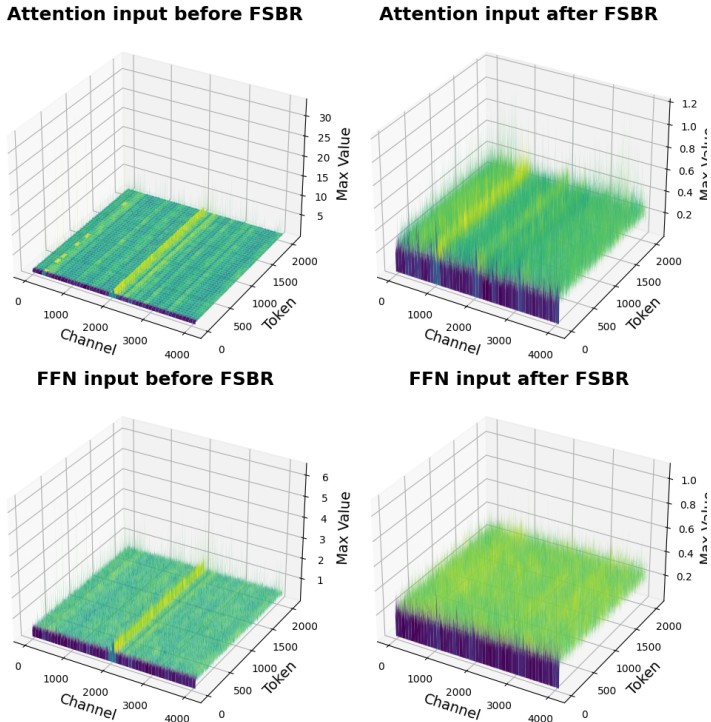

Figure 6: The activation distribution of FFN and Attention of LLaMA-2-7B before and after the FSBR.

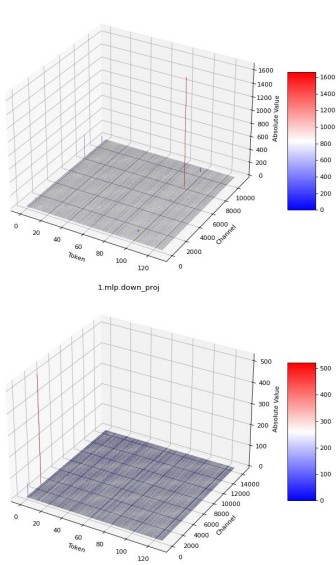

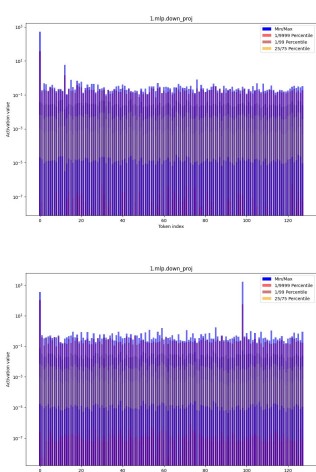

Figure 7: 3D Visualization in activations of LLaMA2 7b(up) LLaMA3 8b(down) model across the token dimensions.

Figure 8: 2D Visualization in activations of LLaMA2 7b(up) LLaMA3 8b(down) model across the token dimensions.

