# OpenReview forum: "I-LLM: Efficient Integer-Only Inference for Fully-Quantized Low-Bit Large Language Models"
_ICLR.cc/2025/Conference — Submitted to ICLR 2025_

### Official Review · Reviewer_Ge8G · 2024-10-28

**Soundness:** 3
**Presentation:** 1
**Contribution:** 2
**Rating:** 3
**Confidence:** 4

**Summary:**

This paper introduces I-LLM, a novel integer-based post-training quantization (PTQ) framework for large language models (LLMs). Traditional PTQ techniques involve a mix of integer and floating-point operations, which limits deployment on edge devices that lack floating-point capabilities. The authors propose three techniques: Full-Smooth Block Reconstruction (FSBR) to smooth activation variations across channels, Dynamic Integer Matrix Multiplication (DI-MatMul) to manage variations between tokens, and dynamic integer implementations for nonlinear operations. Experiments show that I-LLM achieves comparable post-compression performance with existing methods while significantly reducing computational overhead.

**Strengths:**

1. The proposed framework enables integer-only quantization for LLMs: by completely avoiding floating-point operations, I-LLM takes an important step toward making LLMs deployable on edge devices, achieving faster and more efficient inference on hardware without the need for floating-point support.

2. The authors conducted extensive experiments across various LLM architectures, model sizes, and datasets, with the table data from the manuscript demonstrating overall outstanding performance.

3. This paper introduces techniques like FSBR and DI-MatMul that optimize LLM quantization accuracy by addressing variations across channels and tokens. These techniques help maintain high precision during the inference process.

**Weaknesses:**

1. Section 3.2 of the paper proposes 'training a smoothing coefficient for all activations and weights to aid in restoring the model’s quantization accuracy.' However, the training and solving process of this coefficient is not discussed, which could lead to confusion and misunderstandings. If the reason for not detailing this part is that the method aligns with SmoothQuant or OmniQuant, it should be explicitly cited and clearly explained.

2. Equations (1) and (2) extend SmoothQuant by applying smoothing to 'NonLinear Act-Smooth.' However, the motivation of SmoothQuant is to reduce the difficulty of quantization by lowering the smoothness of activations through a smoothing coefficient. In Equations (1) and (2), the smoothing coefficient is counterbalanced between the Gate-layer and Up-layer using '*s' and '/s', respectively. The paper does not discuss the rationale behind this operation or why 'W' scale is 'times' while the 'V' is 'division'.

3. The definition of $\sigma$ in Equation (2) is confusing. Based on Equation (2) and line 262, it follows that $\sigma'(x1) = \sigma(x1 / s)$, and $\sigma'(x1') = \sigma(x1' / s)$. So we can get $\sigma(x1' / s) = \sigma(x1 / s)$, i do believe this is not a right equation or hope the author can make a clarification on this.

4. The authors state in line 270 that 'SmoothQuant and OmniQuant are subsets of FSBR.' However, based on the description in this section, it appears that FSBR actually adopts the techniques of SmoothQuant and OmniQuant and extends them within 'NonLinear Act-Smooth.' Referring to them as subsets is inaccurate and could lead to misunderstandings.

5. I have carefully reviewed and evaluated the anonymous code repository provided by the authors. Regarding the DI-MatMul computation process mentioned in Section 3.3 (specifically in *quantize/quant_modules/QuantMatMul, quantize/quantizer), its implementation and definition of 'dynamic' is consistent with the OmniQuant codebase. If there are any omissions or misunderstandings, I would appreciate further clarification from the authors.

6. Since the quantizer used in the paper results in the same post-quantization weight bit-width and additional quantizer parameters (scaling factor, zero factor) as methods like OmniQuant, the specific factors behind the reduction in **Weight Memory** listed in Table 4 for I-LLM are not clearly discussed. It would be helpful if the authors could clarify which specific parameter compression or operation contributes to this memory efficiency advantage.

7. In table 5, 'OmniquantQuant'->'Omniquant', 'CLippedSoftamx'->'CLippedSoftmax'

**Questions:**

Please refer to Weaknesses

---

> ### Author Response · Authors · 2024-11-29
> **Response to Reviewer Ge8g Part 1**
>
> We sincerely appreciate the time and effort you have dedicated to reviewing our manuscript. We have made every effort to address all your concerns, as detailed below.
>
> > Section 3.2 of the paper proposes 'training a smoothing coefficient for all activations and weights to aid in restoring the model’s quantization accuracy.' However, the training and solving process of this coefficient is not discussed, which could lead to confusion and misunderstandings. If the reason for not detailing this part is that the method aligns with SmoothQuant or OmniQuant, it should be explicitly cited and clearly explained.
>
> Existing LLM quantization methods (e.g., SmoothQuant and OmniQuant) **typically focus only on linear layers**, often neglecting the complex quantization interactions between non-linear layers and linear layers (e.g., MatMul <=> Softmax, Gate Projection <=> SiLU). These interactions, however, can have a significant impact on the performance of integer-only quantization. In contrast, our FSBR explicitly accounts for the quantization interplay between the inputs of each layer and the outputs of the preceding layers, and conducts comprehensive smoothing accross all the layers in LLMs. Furthermore, FSBR ensures that each layer seamlessly accommodates the quantized outputs of the previous layer, eliminating the need for explicit quantization or dequantization steps. This approach enables a more efficient and streamlined computation process. By optimizing data distribution through a holistic block-level inference graph, **FSBR paves the way for efficient integer-only arithmetic inference while preserving high accuracy.**
>
> It is important to emphasize that, even without considering the contribution of integer-only related methods, **the remaining innovations of FSBR still lead to significant accuracy improvements and outperform conventional LLM quantization methods** (as shown in Table 9). These innovations, first introduced in FSBR, include Group Query Attention Optimization, Up Projection , Down Projection Smoothing & Nonlinear Activation Smoothing.
>
> We summarize the unique contributions of FSBR as follows, distinguishing them from existing methods such as SmoothQuant, OminiQuant and Spinquant:
> - **Linear Smoothing**
>
>     - **Norm-Linear Smoothing**: Illustrated in Figure 5 (Method 2) and inspired by SmoothQuant, I-LLM incorporates smoothing coefficients between normalization (Norm) layers and linear layers. This reduces the inter-channel differences in activations and weights in QKV projections and Up/Gate operations effectively.
>     - **QK Smoothing**:  As shown in Figure 5 (Method 1), inter-channel discrepancies frequently occur in Query and Key matrix computations. FSBR addresses this using an equivalent transformation: $$\boldsymbol{XW} _q\boldsymbol{W} _k^\top \boldsymbol{X} =\boldsymbol{X}(\boldsymbol{W} _q\cdot \boldsymbol{S} _{qk})(\boldsymbol{W} _k/\boldsymbol{S} _{qk})$$ which jointly smooths the activations of Query and Key matrices, balancing the inter-channel differences.
>     - **OV Smoothing**: Depicted in Figure 5 (Method 3), FSBR introduces head-wise smoothing between Value Projection and Output Projection layers. For each attention head, the interaction is expressed as:
>          $$\boldsymbol{P}\boldsymbol{x}^{h} _v\boldsymbol{W}^{h} _v\boldsymbol{W}^{h} _o = \boldsymbol{P}\boldsymbol{x}^h _v\boldsymbol{W}^h _v\boldsymbol{S}^h _{ov}\boldsymbol{S}^{-1^h} _{ov}\boldsymbol{W}^h _o,$$
>
>         where $\boldsymbol{P}$ represents the attention mechanism and $\boldsymbol{S}_{ov}$ is a learnable per-channel scaling
>         matrix between Output Projection and Value Projection. This approach smooths the inter-channel differences of the Value
>         Projection’s output activations and the Output Projection’s input activations simultaneously.
>   - **Up Down Smoothing**: Also depicted in Figure 5 (Method 3), FSBR introduces smoothing coefficients between Up and Down Projections based on computational invariance:
>     $$\boldsymbol{x} _{up}\boldsymbol{W} _{up}\otimes\boldsymbol{x} _{down}\boldsymbol{W} _{down} = \boldsymbol{x} _{up}\boldsymbol{W} _{up}\boldsymbol{S} _{u|d}\otimes\boldsymbol{x} _{down} \boldsymbol{W} _{down} \boldsymbol{S} _{u|d} ^{-1}$$
>
>     where $\boldsymbol{S}_{u|d}$ is a learnable per-channel smoothing coefficients between Up Projection and Down Projection. This
>     method effectively reduces inter-channel differences between the Up Projection’s output activations and the Down Projection’s
>     input activations.

---

> ### Author Response · Authors · 2024-11-29
> **Response to Reviewer Ge8g Part 2**
>
> - **Nonlinear Activation Optimization**
>     1. **Channel-wise Smoothing for SiLU Activations:**
>
>         For the first time, FSBR proposes the NonLinear Act-Smooth method, as depicted in Figure 5 (Method 4), to smooth inter-channel differences in SiLU activations. Specifically, a smoothing parameter $s$ is introduced, enabling inputs $x_1$ and $x_2$ to be equivalently reconstructed into a more quantization-friendly form (see Equation 2). This innovation significantly reduces the quantization error of SwiGLU activations (see Figure 4).
>
>     2. **Softmax Input Smoothing:**
>
>         The issue of outliers in Softmax activations is particularly pronounced. FSBR addresses this by leveraging the underflow characteristics of the exponential function. Using dynamic truncation, the input range of Softmax is adjusted to lie within $(-c, 0)$ (see Equation 11), effectively limiting the quantization range and reducing input quantization errors.
>
>
> The detailed training procedure is as follows:
>
> 1. **Collection of Activation Magnitudes:** We begin by using a subset of the calibration set (e.g., 128 samples of 2048-token sequences) to collect the activation magnitudes for each channel in every layer, denoted as $\boldsymbol{x}_{max}$, and store the values.
>
> 2. **Preparation of Input Data:** The initial input token data for the first block is derived from the embedding layer using the calibration set and is denoted as $\boldsymbol{X}$.
>
> 3. **Block-wise Quantization:** For the i-th block:
>
>     1. **Floating-point Output Calculation:** Compute the floating-point output of the block ,  $\boldsymbol{Y}$,using $\boldsymbol{X}$. This serves as the reference for loss computation.
>     2. **Pseudo-quantization Replacement:** Replace the linear layers, activation layers, Softmax layers, and normalization layers with their pseudo-quantized versions. Generate the smoothing parameters as shown in Figure 5, and compute the initial value of the smoothing parameter using the activation and weight magnitudes saved in step one, based on the formula $S=(\boldsymbol{x} _{max}/ \boldsymbol{w} _{max})^{1/2}$.
>     3. **Optimization:** Construct an optimizer to perform block-wise reconstruction, optimizing all smoothing parameters and weight clipping factors.
>     4. **Smoothing Parameter Integration:** Incorporate the optimized smoothing parameters into the corresponding weights and pass $\boldsymbol{X}$ through the quantized version of the current block to generate the updated $\boldsymbol{X}$.
>     5. **Integer-only Conversion:** Convert the current block into an integer-only version.
> 4. **Quantization Completion:** After processing all blocks, the quantized model is finalized and evaluated.

---

> ### Author Response · Authors · 2024-11-29
> **Response to Reviewer Ge8g Part 3**
>
> > Equations (1) and (2) extend SmoothQuant by applying smoothing to 'NonLinear Act - Smooth.' However, the motivation of SmoothQuant is to reduce the difficulty of quantization by lowering the smoothness of activations through a smoothing coefficient. In Equations (1) and (2), the smoothing coefficient is counterbalanced between the Gate - layer and Up - layer using '*s' and '/s', respectively. The paper does not discuss the rationale behind this operation or why 'W' scale is 'times' while the 'V' is 'division'.
>
> The symbols "*s" and "/s" are used to mantain computational equivalence, resembling the operation $x_1\otimes x_2 =x_1\otimes s \oslash  x_2$ Here, multiplication($\otimes$) and division($\oslash$) are relative operations depending on whether $s>1$or $s<1$. In this context, $x_1$and $x_2$represents  the outputs of the Up projection and the gating mechanism in SwiGLU, respectively. We elaborate on the underlying principles below:
>
>
> $$\operatorname{SwiGLU}(\boldsymbol{x}, \boldsymbol{W}, \boldsymbol{V}, \boldsymbol{b}, \boldsymbol{c}) $$
>
> $$ = \operatorname{SiLU}(\boldsymbol{x}\boldsymbol{W} + \boldsymbol{b}) \otimes (\boldsymbol{x}\boldsymbol{V} + \boldsymbol{c}) $$
> $$ = (\boldsymbol{x}\boldsymbol{W} + \boldsymbol{b}) \otimes (\boldsymbol{x}\boldsymbol{V} + \boldsymbol{c}) \otimes \sigma(\boldsymbol{x}\boldsymbol{W} + \boldsymbol{b}) $$
> $$= [(\boldsymbol{x}\boldsymbol{W} + \boldsymbol{b}) \otimes \boldsymbol{s}] \otimes [(\boldsymbol{x}\boldsymbol{V} + \boldsymbol{c}) \oslash \boldsymbol{s}] \otimes  \sigma([(\boldsymbol{x}\boldsymbol{W} + \boldsymbol{b}) \otimes \boldsymbol{s}] \oslash \boldsymbol{s})
> $$
>
> Let $\boldsymbol{W}' = \boldsymbol{W}\otimes \boldsymbol{s}, $
> $\boldsymbol{b}' = \boldsymbol{b}\otimes \boldsymbol{s}, \boldsymbol{V}' = \boldsymbol{V}\oslash \boldsymbol{s}, \boldsymbol{c}' = \boldsymbol{c}\oslash \boldsymbol{s}, \boldsymbol{x1'}=\boldsymbol{x}\boldsymbol{W}'+\boldsymbol{b'}, \boldsymbol{x2'} = \boldsymbol{x}\boldsymbol{V}' + \boldsymbol{c'}, \sigma'(\boldsymbol{x1'}) = \sigma(\boldsymbol{x1'}\oslash\boldsymbol{s})$
>
> We can derive that
>
> $\begin{aligned}
> \operatorname{SwiGLU}(\boldsymbol{x}, \boldsymbol{W}, \boldsymbol{V}, \boldsymbol{b}, \boldsymbol{c}) = (\boldsymbol{x1}\otimes \boldsymbol{s})\otimes (\boldsymbol{x2}\oslash \boldsymbol{s}) \otimes \sigma'(\boldsymbol{x1})
> = \boldsymbol{x1'} \otimes \boldsymbol{x2'} \otimes{\sigma '}(\boldsymbol{x1'})
> \end{aligned}$
>
> > The definition of σ in Equation (2) is confusing. Based on Equation (2) and line 262, it follows that $σ ′ (x 1) = σ (x 1 / s )$, and $σ ′ (x 1 ′ ) = σ (x 1 ′ / s )$ So we can get $σ(x 1 ′ / s) = σ (x 1 / s )$, i do believe this is not a right equation or hope the author can make a clarification on this.
>
> We would like to clarify that, given  $σ ′ (x 1) = σ (x 1 / s )$,  it naturally follows that  $\sigma'(x1')= \sigma(x1'/s)$.
>
> To further address your concern, we also followed your line of reasoning and derived the following:
>
> $σ (x 1 ′ / s ) = σ ′ (x 1 ′ )$, $σ (x 1 / s ) = σ ′ (x 1)$, so $σ(x 1 ′ / s) \not= σ (x 1 / s )$

---

> ### Author Response · Authors · 2024-11-29
> **Response to Reviewer Ge8g Part 4**
>
> > The authors state in line 270 that 'SmoothQuant and OmniQuant are subsets of FSBR.' However, based on the description in this section, it appears that FSBR actually adopts the techniques of SmoothQuant and OmniQuant and extends them within 'NonLinear Act - Smooth.' Referring to them as subsets is inaccurate and could lead to misunderstandings.
>
> Existing LLM quantization methods (e.g., SmoothQuant and OmniQuant) **typically focus only on linear layers**, often neglecting the complex quantization interactions between non-linear layers and linear layers (e.g., MatMul <=> Softmax, Gate Projection <=> SiLU). These interactions, however, can have a significant impact on the performance of integer-only quantization. In contrast, our FSBR explicitly accounts for the quantization interplay between the inputs of each layer and the outputs of the preceding layers, and conducts comprehensive smoothing accross all the layers in LLMs. Furthermore, FSBR ensures that each layer seamlessly accommodates the quantized outputs of the previous layer, eliminating the need for explicit quantization or dequantization steps. This approach enables a more efficient and streamlined computation process. By optimizing data distribution through a holistic block-level inference graph, **FSBR paves the way for efficient integer-only arithmetic inference while preserving high accuracy.**
>
> It is important to emphasize that, even without considering the contribution of integer-only related methods, **the remaining innovations of FSBR still lead to significant accuracy improvements and outperform conventional LLM quantization methods** (as shown in Table 9). These innovations, first introduced in FSBR, include Group Query Attention Optimization, Up Projection , Down Projection Smoothing & Nonlinear Activation Smoothing.
>
> We summarize the unique contributions of FSBR as follows, distinguishing them from existing methods such as SmoothQuant, OminiQuant and Spinquant:
>
> 1. **Linear Smoothing**
>     - **Group Query Attention Optimization:**
>
>         As illustrated in Figure 5 (Method 3), FSBR performs head-wise smoothing between the Value Projection and Output Projection layers. For each attention head, the interaction between these layers is expressed as:
>
>         $$\boldsymbol{P}\boldsymbol{x}^{h} _v\boldsymbol{W}^{h} _v\boldsymbol{W}^{h} _o = \boldsymbol{P}\boldsymbol{x}^h _v\boldsymbol{W}^h_v\boldsymbol{S}^h _{ov}\boldsymbol{S}^{-1^h} _{ov}\boldsymbol{W}^h _o,$$
>
>         where $\boldsymbol{P}$ represents the attention mechanism and $\boldsymbol{S}_{ov}$ is a learnable per-channel scaling matrix between Output Projection and Value Projection. This approach smooths the inter-channel differences of the Value Projection’s output activations and the Output Projection’s input activations simultaneously.
>
>     - **Up Projection and Down Projection Smoothing:**
>
>         Also shown in Figure 5 (Method 3), FSBR introduces smoothing coefficients between Up and Down Projections based on computational invariance:
>
>         $$\boldsymbol{x} _{up}\boldsymbol{W} _{up}\otimes\boldsymbol{x} _{down}\boldsymbol{W} _{down} = \boldsymbol{x} _{up}\boldsymbol{W} _{up} \boldsymbol{S} _{u|d}\otimes\boldsymbol{x} _{down}\boldsymbol{W} _{down}\boldsymbol{S} _{u|d}^{-1},$$
>
>         where $\boldsymbol{S}_{u|d}$ is a learnable per-channel smoothing coefficients between Up Projection and Down Projection. This method effectively reduces inter-channel differences between the Up Projection’s output activations and the Down Projection’s input activations.
> 2. **Nonlinear Activation Optimization**
>     - **Channel-wise Smoothing for SiLU Activations:**
>
>         For the first time, FSBR proposes the NonLinear Act-Smooth method, as depicted in Figure 5 (Method 4), to smooth inter-channel differences in SiLU activations. Specifically, a smoothing parameter $s$ is introduced, enabling inputs $x_1$ and $x_2$ to be equivalently reconstructed into a more quantization-friendly form (see Equation 2). This innovation significantly reduces the quantization error of SwiGLU activations (see Figure 4).
>
>     - **Softmax Input Smoothing:**
>
>         The issue of outliers in Softmax activations is particularly pronounced. FSBR addresses this by leveraging the underflow characteristics of the exponential function. Using dynamic truncation, the input range of Softmax is adjusted to lie within $(-c, 0)$ (see Equation 11), effectively limiting the quantization range and reducing input quantization errors.
>
> - Our method is not merely an extension but fully leverages the block structure of LLMs to **smooth inter-channel differences for all suitable activation-activation and activation-weight pairs. This effectively mitigates the performance degradation typically caused by full quantization.**
>
> - Even under fully integer quantization configurations, I-LLM outperforms non-integer methods such as SmoothQuant and OmniQuant, as demonstrated in Table 2 and Table 3 of the paper.

---

> ### Author Response · Authors · 2024-11-29
> **Response to Reviewer Ge8g Part 5**
>
> > I have carefully reviewed and evaluated the anonymous code repository provided by the authors. Regarding the DI - MatMul computation process mentioned in Section 3.3 (specifically in quantize/quant_modules/QuantMatMul, quantize/quantizer), its implementation and definition of 'dynamic' is consistent with the OmniQuant codebase. If there are any omissions or misunderstandings, I would appreciate further clarification from the authors.
>
> In the anonymous repository, line 339 of quantize/FSBR.py states: "if args.illm: TODO We will soon open source implementations of these integer-only operators." We have clearly indicated that the implementation of these integer-only operators has not yet been open-sourced. As we are a chip company, this implementation is part of our core intellectual property and a key competitive advantage. According to company policy, the kernel implementation can only be fully disclosed after the paper is officially accepted.
>
> The methods and formulas provided in our paper are fully reproducible. We have already received numerous acknowledgments of successful replication. For instance, ABQLLM [https://arxiv.org/abs/2408.08554] has successfully reproduced our accuracy results.
>
> If it would assist your review process, we are willing to apply for company approval to release this implementation ahead of schedule.
>
> > Since the quantizer used in the paper results in the same post - quantization weight bit - width and additional quantizer parameters (scaling factor, zero factor) as methods like OmniQuant, the specific factors behind the reduction in Weight Memory listed in Table 4 for I - LLM are not clearly discussed. It would be helpful if the authors could clarify which specific parameter compression or operation contributes to this memory efficiency advantage.
>
> Thank you for pointing this out. We have corrected the erroneous information in the table in the revised version. The weight-memory calculation previously did not account for the embedding and head layers.
>
> > In table 5, 'OmniquantQuant'->'Omniquant', 'CLippedSoftamx'->'CLippedSoftmax'
>
> Thank you for pointing out some typo issues. We have addressed them in the revised manuscript.

---

> ### Author Response · Authors · 2024-12-02
> **Title: Eager for Your Feedback and Open to Further Discussion**
>
> Dear Reviewer Ge8G,
>
> We hope you are doing well. We sincerely appreciate your thorough review and constructive criticisms of our paper. Your feedback has been invaluable in helping us improve our work. We have addressed each of your concerns in detail.
>
> Specifically, we have:
>
> * **Detailed the Training and Solving Process of Smoothing Coefficients**: We provided an in-depth explanation of how the smoothing coefficients are learned during the post-training quantization process, clarifying that our method is distinct from SmoothQuant and OmniQuant.
>
> * **Clarified Equations and Operations**: We elaborated on the rationale behind the operations in Equations (1) and (2), explaining why certain scaling factors are applied as multiplication or division, and provided mathematical justifications to clarify any confusion.
>
> * **Addressed Definitions and Notations**: We corrected and clarified the definitions in Equation (2) related to the SiLU function and the smoothing coefficient to ensure accurate understanding.
>
> * **Revised Statements for Accuracy**: We acknowledged the inaccuracy in stating that SmoothQuant and OmniQuant are subsets of FSBR and have revised the manuscript to more accurately reflect the relationship between these methods.
>
> * **Clarified Implementation Details**: We explained the differences between our DI-MatMul implementation and that of OmniQuant, highlighting the novel aspects of our integer-only operators.
>
> * **Corrected Table Errors**: We addressed the issues you pointed out in Table 5 and corrected any typographical errors in the manuscript.
>
> We invite you to review our revised manuscript and responses. We are eager to hear your thoughts and welcome any further feedback or discussion you may have. Your insights have been crucial in refining our work.
>
> Thank you for your time and valuable contributions.
>
> Best regards,
>
> The Authors

---

### Official Review · Reviewer_Hjwd · 2024-11-02

**Soundness:** 3
**Presentation:** 3
**Contribution:** 3
**Rating:** 6
**Confidence:** 3

**Summary:**

This study introduces a novel quantization approach for large language models (LLMs), featuring custom-designed modules: DI-MatMul, DI-ClippedSoftmax, DI-Exp, and DI-Normalization. These modules collectively outperform state-of-the-art (SOTA) methods, offering enhanced performance in LLM applications.

**Strengths:**

The authors provide valuable insights into the quantization of large language models (LLMs), demonstrating how such models can maintain high accuracy despite the reduced precision.

**Weaknesses:**

Consider combining quantization with pruning to achieve enhanced model efficiency and reduced computational overhead.
Consider a hybrid quantization approach, where different layers utilize varied precision levels, such as W4A4 for certain layers and W8A8 for others, to balance efficiency and performance.

**Questions:**

NA

---

> ### Author Response · Authors · 2024-11-29
> **Response for Reviewer Hjwd**
>
> We sincerely appreciate the time and effort you have dedicated to reviewing our manuscript. We have made every effort to address all your concerns, as detailed below.
>
> > Consider combining quantization with pruning to achieve enhanced model efficiency and reduced computational overhead. Consider a hybrid quantization approach, where different layers utilize varied precision levels, such as W4A4 for certain layers and W8A8 for others, to balance efficiency and performance.
>
> We sincerely appreciate your insightful suggestions on integrating quantization with pruning and exploring hybrid quantization with varied precision levels across layers. I-LLM focuses on fully quantized post-training quantization (PTQ) algorithms and integer-only inference. Currently, I-LLM is decoupled from pruning and mixed-precision approaches, making the integration of these methods a promising direction. Inspired by your recommendations, we will consider the following areas in our future research:
> - Combining pruning with quantization, with a focus on maintaining activation stability to complement our integer-only quantization techniques.
> - Exploring hybrid-precision quantization to optimize the trade-off between computational cost and model performance, particularly for hardware platforms that support mixed-precision computation.
> Thank you again for these constructive suggestions. We believe they provide valuable insights to guide the next phase of our research.

---

### Official Review · Reviewer_UDtp · 2024-11-03

**Soundness:** 4
**Presentation:** 3
**Contribution:** 3
**Rating:** 6
**Confidence:** 4

**Summary:**

The paper present an end-to-end quantization method that is applied to all the inference sub-steps in LLMs, including non-linear operations and all attention calculations. The technique presented leverages block and dynamic quantization concepts. Specific integer approximation for non-linearities are presented to avoid moving o fp when computing them, End-to-end results are presented, and an implementation of the method on GPU is also presented, showing significant speedups.

**Strengths:**

First method that covers all the steps of integer inference.
Presents implementation results, showing significant speedup even in non-adhoc designed hardware
end-to-end results in inference are presented, not only sub-steps

**Weaknesses:**

It seems the method is not post-training (the smoothing coefficients are learned), hence it is not applicable for the largest LLMs for which training is not really easy to repeat.
Notation is hard to follow and could be simplified to get better intuition on the methods applied.
Approximation proposed for non-linear functions are not really well justified
Similar ideas are present in I-BERT, I-ViT, BRECQ. Thus, the contribution of this paper seems to be incremental (see BRECQ: Pushing the Limit of Post-Training Quantization by Block Reconstruction)

**Questions:**

Detail more in depth the steps needed in training.

---

> ### Author Response · Authors · 2024-11-29
> **Response to Reviewer UDtp Part 1**
>
> We sincerely appreciate the time and effort you have dedicated to reviewing our manuscript. We have made every effort to address all your concerns, as detailed below.
>
> > It seems the method is not post - training (the smoothing coefficients are learned), hence it is not applicable for the largest LLMs for which training is not really easy to repeat.
>
> Our method is a post-training quantization approach that does not require weight retraining and only uses a small calibration set (128 samples), consistent with GPTQ.
>
> This method has low computational requirements and can quantize a 70B model within 16 hours using a single A6000 GPU. Even consumer-grade GPUs like the 3090 can effectively quantize 30B LLMs.
>
> > Notation is hard to follow and could be simplified to get better intuition on the methods applied.
>
> Thank you for your suggestion. We will strive to simplify the formula expressions to make them easier to understand.
>
> In our manuscript we use the superscript $I$ to denote integers, such as $x^I$. Lowercase symbols represent scalars,  bold uppercase symbols like $\boldsymbol{X}$ represent tensors, and bold lowercase symbols like $\boldsymbol{x}$ represent vectors.
>
> In the actual inference process of I-LLM, only integer operations—denoted by the superscript $I$—are involved, as shown in algorithms of Appendix A.3 and Equation . Any non-integer floating-point computations are included solely for conceptual clarity.
>
> > Approximation proposed for non - linear functions are not really well justified Similar ideas are present in I - BERT, I - ViT, BRECQ. Thus, the contribution of this paper seems to be incremental (see BRECQ: Pushing the Limit of Post - Training Quantization by Block Reconstruction)
>
> In Section 3.4.1, we provide detailed proof of DI-Softmax and DI-Exp. The critical aspect of the fully integer implementation of the Norm function lies in calculating the square root, which we achieve using the bitwise comparison method outlined in Algorithm 4.  The nonlinearity of the SiLU function originates from the sigmoid function, where $\text{sigmoid}(x)=\frac{1}{1+e^{-x}}=\frac{e^x}{e^x+1}$, and $e^x$ can be approximated using DI-Exp.
>
> I-BERT and I-ViT are excellent algorithms for BERT and ViT, respectively. However, they are specifically designed for BERT and Vision Transformers. When applied directly to large language models (LLMs), as shown in Figure 2, they suffer significant accuracy degradation.
>
> BRECQ determines the rounding direction for weight quantization and the step size for activation quantization through block-wise reconstruction. While suitable for smaller models, applying BRECQ to LLMs introduces a large number of trainable parameters, leading to substantial optimization costs—potentially hundreds of hours of training time. Moreover, BRECQ cannot address the issue of inter-channel variance in activations. This is fundamentally different from the Fully-Smooth Block Reconstruction (FSBR) method proposed in our paper. FSBR reduces inter-channel variance of all activations and weights within a block using learnable smoothing coefficients, thereby easing the quantization process. Additionally, FSBR introduces learnable weight clipping to mitigate the impact of weight outliers, effectively reducing the dynamic range of weights and improving quantization accuracy.
>
> Traditional integer-only inference methods, such as I-ViT and I-BERT, have primarily targeted small language and vision models, neglecting the unique challenges posed by LLMs. As demonstrated in Figure 3, LLMs exhibit a wide dynamic range of activations alongside significant inter-channel and inter-token variability, making these traditional methods ineffective even under 8-bit quantization (Figure 2). To address this gap, I-LLM introduces practical integer-only inference for LLMs, explicitly accommodating their distinct activation distribution characteristics through a novel integer-only dynamic quantization strategy. Tailored implementations for key operators—including MatMul, LayerNorm, Softmax, and the SiLU activation—combine innovative dynamic designs and efficient nonlinear approximations. These operators enable accurate low-bit computation while maintaining floating-point-level performance.

---

> ### Author Response · Authors · 2024-11-29
> **Response to Reviewer UDtp Part 2**
>
> Below, we detail the implementation and benefits of each operator:
> - **Dynamic Integer-only Matrix Multiplication (DI-MatMul):**
>
>     DI-MatMul overcomes the limitations of static quantization, which fails to adapt to varying activation ranges, by employing a dynamic quantization strategy with integer scale and zero-point (zp) processing. In DI-MatMul, quantization scales and zero-points for inputs and outputs are dynamically adjusted, significantly reducing quantization errors. Unlike methods such as I-ViT that use fixed parameters, DI-MatMul relies on integer arithmetic for dynamic quantization, as detailed in Equations (4)-(7). **Intermediate activations undergo dynamic quantization through integer right shifts and multiple adjustments, ensuring consistent output precision across diverse input ranges.** Moreover, DI-MatMul eliminates the need for dynamic quantization of inputs and dequantization of outputs in both weight-activation and activation-activation quantization scenarios, achieving a 1.2× speedup for the former and 2.1× for the latter (Table 7 in ours manuscript).
>
> - **Dynamic Integer-only Exponential Function (DI-Exp):**
>
>     Both I-ViT and DI-Exp employ the logarithmic base-change formula and linear interpolation for exponential calculations. However, DI-Exp introduces significant enhancements:
>
>     - **Support for Dynamic Quantization:** Unlike I-ViT, which assumes static quantization, DI-Exp accommodates dynamically quantized inputs, ensuring compatibility with integer-only workflows.
>
>     - **Improved Representation of $log_2e$.** DI-Exp replaces I-ViT’s approximate representation($1.0111b$) with a more precise formula $m^I_e >> k^I_e$, as shown in Algorithm 1. This approach eliminates unnecessary shifts, refines precision, and avoids division by substituting bitwise shifts for more accurate quotients($\boldsymbol{p}^I$) and remainders($\boldsymbol{r}^I$). Experimental results show DI-Exp achieves over 5× speedup compared to FP16 with negligible accuracy loss for Softmax and SiLU (Tables 5 and 8 in manuscript).
>
> - **Dynamic Integer-only Softmax (DI-ClippedSoftmax):**
>
>     Softmax values often span a wide dynamic range, especially with increasing tokens, leading to significant quantization errors (e.g., 8-bit quantization in I-ViT yields an error of 0.15 for ranges like [-20, 20], Figure 3). DI-ClippedSoftmax mitigates this by dynamically truncating activation ranges (e.g., [xmax − 15, xmax]), leveraging the exponential function’s underflow properties. This strategy ensures an input activation error below 0.03 with no additional accuracy loss (Table 5).
>
> - **Dynamic Integer-only Normalization (DI-Norm) and Dynamic Integer-only SwiGLU (DI-SwiGLU):**
>
>     LayerNorm and RMSNorm are essential in Transformers, with RMSNorm exhibiting greater activation variability across channels. I-LLM applies per-channel dynamic quantization to these normalization inputs, combined with channel-wise smoothing via FSBR, to significantly reduce inter-channel quantization errors. Unlike traditional methods (e.g., Newton’s method in I-BERT), DI-Norm employs bitwise comparison during inference for higher precision and consistency. Additionally, I-LLM addresses the challenge of quantizing SwiGLU—complicated by inter-channel activation differences—by decomposing SiLU into linear and nonlinear components. Through dynamic quantization and DI-Exp, I-LLM resolves these discrepancies effectively (Section 3.2, Algorithm 3).

---

> ### Author Response · Authors · 2024-11-29
> **Response to Reviewer UDtp Part 3**
>
> > Detail more in depth the steps needed in training.
>
> The detailed training procedure is as follows:
>
> 1. **Collection of Activation Magnitudes:** We begin by using a subset of the calibration set (e.g., 128 samples of 2048-token sequences) to collect the activation magnitudes for each channel in every layer, denoted as $\boldsymbol{x}_{max}$, and store the values.
> 2. **Preparation of Input Data:** The initial input token data for the first block is derived from the embedding layer using the calibration set and is denoted as $\boldsymbol{X}$.
> 3. **Block-wise Quantization:** For the i-th block:
>
>     1. **Floating-point Output Calculation:** Compute the floating-point output of the block ,  $\boldsymbol{Y}$,using $\boldsymbol{X}$. This serves as the reference for loss computation.
>
>     2. **Pseudo-quantization Replacement:** Replace the linear layers, activation layers, Softmax layers, and normalization layers with their pseudo-quantized versions. Generate the smoothing parameters as shown in Figure 5, and compute the initial value of the smoothing parameter using the activation and weight magnitudes saved in step one, based on the formula $S=(\boldsymbol{x} _{max}/\boldsymbol{w} _{max})^{1/2}$.
>
>     3. **Optimization:** Construct an optimizer to perform block-wise reconstruction, optimizing all smoothing parameters and weight clipping factors.
>
>     4. **Smoothing Parameter Integration:** Incorporate the optimized smoothing parameters into the corresponding weights and pass $\boldsymbol{X}$ through the quantized version of the current block to generate the updated $\boldsymbol{X}$.
>
>     5. **Integer-only Conversion:** Convert the current block into an integer-only version.
>
> 4. **Quantization Completion:** After processing all blocks, the quantized model is finalized and evaluated.

---

> ### Author Response · Authors · 2024-12-02
> **Eager to Hear Your Feedback and Continue the Discussion**
>
> **Dear Reviewer UDtp,**
>
> We hope you are doing well. We are grateful for your positive assessment of our work and for your thoughtful suggestions. We have taken your feedback into careful consideration and have provided detailed explanations in our response.
>
> Specifically, we have:
>
> • **Clarified Post-Training Nature of Our Method**: We emphasized that our approach is a post-training quantization method that does not require weight retraining. We detailed how the smoothing coefficients are learned using a small calibration set, making it applicable even to large LLMs without extensive retraining.
>
> • **Simplified Notations for Clarity**: We acknowledged that some notations were hard to follow and have revised the manuscript to simplify expressions and enhance the intuitive understanding of our methods.
>
> • **Justified Non-Linear Function Approximations**: We provided more in-depth explanations and proofs for our integer-only implementations of non-linear functions like Softmax and SiLU, highlighting how they differ from previous methods like I-BERT and I-ViT.
>
> • **Detailed Training Steps**: We included a comprehensive description of the steps involved in training, detailing the collection of activation magnitudes, block-wise quantization process, and optimization of smoothing parameters.
>
> We are keen to hear your thoughts on our revisions and would greatly appreciate any further feedback or discussion. Your insights are invaluable in enhancing the clarity and impact of our work.
>
> Thank you for your time and support.
>
> Sincerely,
>
> The Authors

---

### Official Review · Reviewer_oGix · 2024-11-04

**Soundness:** 2
**Presentation:** 2
**Contribution:** 2
**Rating:** 5
**Confidence:** 4

**Summary:**

The paper proposes I-LLM, an integer-only post-training quantization (PTQ) framework for large language models (LLMs), aiming to improve inference efficiency on hardware lacking floating-point support. The key contributions include the Fully-Smooth Block-Reconstruction (FSBR) to smooth inter-channel variations in activation and dynamic quantization methods (Dynamic Integer-only MatMul and others). Experiments on several LLMs, such as LLaMA and OPT, demonstrate improved modest speed and memory usage under W4A4 settings, with minimal accuracy loss compared to floating-point (FP) models. Despite these claims, the methodology largely extends known techniques, showing limited novelty, and lacks sufficient evaluation against state-of-the-art (SOTA) approaches on challenging quantization settings.

**Strengths:**

- The integer-only quantization of non-linear operations offers potential efficiency improvements for hardware without FP support.
- Demonstrates notable performance in W4A4 quantization settings, showing significant reductions in latency and memory usage on specific LLMs.

**Weaknesses:**

- The proposed techniques, such as FSBR, heavily build on prior works like OmniQuant for quantization-parameter tuning and I-VIT for dynamic quantization, limiting originality.
- The absence of comprehensive SOTA comparison limits the rigor of performance claims, particularly missing comparisons with rotation-based methods (e.g., SpinQuant (Liu et al., 2024)) and LUT-based approximations (e.g., NN-LUT(Yu et al., 2022)).
- FSBR and DI-MatMul introduce computational overhead with on-the-fly operations like 8-bit scalar multiplication/division, yet no detailed ablation study quantifies the latency impact per Transformer component.
- The evaluation datasets and tasks are limited, and broader testing across more diverse and challenging benchmarks is required to substantiate the generalization of results.

**Questions:**

- What framework and baseline setup were used for W4A4 quantization, and can more detail be provided about the experimental environment?
- The description of FSBR is unclear. Could the authors provide a more explicit breakdown of its application across different computing units beyond MatMul?
- The authors also emphasize the outliers across the token (which sounds new), but Fig. 3 shows that the token-wise distribution looks much flatter than the channel-wise distribution, which is well known.

---

> ### Author Response · Authors · 2024-11-29
> **Response to Reviewer oGix Part 1**
>
> We sincerely appreciate the time and effort you have dedicated to reviewing our manuscript. We have made every effort to address all your concerns, as detailed below.
>
> >The proposed techniques, such as FSBR, heavily build on prior works like OmniQuant for quantization - parameter tuning and I - VIT for dynamic quantization, limiting originality.
>
> We will address your question from two perspectives: LLM quantization and Integer-only quantization.
>
> ---
>
> **LLM quantization**
>
> Existing LLM quantization methods (e.g., SmoothQuant[1] and OmniQuant[2]) **typically focus only on linear layers**, often neglecting the complex quantization interactions between non-linear layers and linear layers (e.g., MatMul <=> Softmax, Gate Projection <=> SiLU). These interactions, however, can have a significant impact on the performance of integer-only quantization. In contrast, our FSBR explicitly accounts for the quantization interplay between the inputs of each layer and the outputs of the preceding layers, and conducts comprehensive smoothing accross all the layers in LLMs. Furthermore, FSBR ensures that each layer seamlessly accommodates the quantized outputs of the previous layer, eliminating the need for explicit quantization or dequantization steps. This approach enables a more efficient and streamlined computation process. By optimizing data distribution through a holistic block-level inference graph, **FSBR paves the way for efficient integer-only arithmetic inference while preserving high accuracy.**
>
> It is important to emphasize that, even without considering the contribution of integer-only related methods, **the remaining innovations of FSBR still lead to significant accuracy improvements and outperform conventional LLM quantization methods** (as shown in Table 9). These innovations, first introduced in FSBR, include Group Query Attention Optimization, Up Projection , Down Projection Smoothing & Nonlinear Activation Smoothing.
>
> We summarize the unique contributions of FSBR as follows, distinguishing them from existing methods such as SmoothQuant, OminiQuant and Spinquant:
>
> - **Linear Smoothing**
>     -  **Group Query Attention Optimization:**
>       As illustrated in Figure 5 (Method 3), FSBR performs head-wise smoothing between the Value Projection and Output Projection layers. For each attention head, the interaction between these layers is expressed as:
>
>           $\boldsymbol{P}\boldsymbol{x}^{h} _v\boldsymbol{W}^{h} _v\boldsymbol{W}^{h} _o = \boldsymbol{P}\boldsymbol{x}^h _v\boldsymbol{W}^h _v\boldsymbol{S}^h _{ov}\boldsymbol{S}^{-1^h} _{ov}\boldsymbol{W}^h _o,$
>
>        where $\boldsymbol{P}$ represents the attention mechanism and $\boldsymbol{S}_{ov}$ is a learnable per-channel scaling matrix between Output Projection and Value Projection. This approach smooths the inter-channel differences of the Value Projection’s output activations and the Output Projection’s input activations simultaneously.
>
>     - **Up Projection and Down Projection Smoothing:**
>       Also shown in Figure 5 (Method 3), FSBR introduces smoothing coefficients between Up and Down Projections based on computational invariance:
>
>         $$\boldsymbol{x} _{up}\boldsymbol{W} _{up}\otimes\boldsymbol{x} _{down}\boldsymbol{W} _{down} = \boldsymbol{x} _{up}\boldsymbol{W} _{up} \boldsymbol{S} _{u|d}\otimes\boldsymbol{x} _{down}\boldsymbol{W} _{down}\boldsymbol{S} _{u|d}^{-1},$$
>
>         where $\boldsymbol{S}_{u|d}$ is a learnable per-channel smoothing coefficients between Up Projection and Down Projection. This method effectively reduces inter-channel differences between the Up Projection’s output activations and the Down Projection’s input activations.
>
> - **Nonlinear Activation Smoothing:**
>     -  **Channel-wise Smoothing for SiLU Activations:**
>         For the first time, FSBR proposes the NonLinear Act-Smooth method, as depicted in Figure 5 (Method 4), to smooth inter-channel differences in SiLU activations. Specifically, a smoothing parameter $s$ is introduced, enabling inputs $x_1$ and $x_2$ to be equivalently reconstructed into a more quantization-friendly form (see Equation 2). This innovation significantly reduces the quantization error of SwiGLU activations (see Figure 4).
>     - **Softmax Input Smoothing:**
>       The issue of outliers in Softmax activations is particularly pronounced. FSBR addresses this by leveraging the underflow characteristics of the exponential function. Using dynamic truncation, the input range of Softmax is adjusted to lie within $(-c, 0)$ (see Equation 11), effectively limiting the quantization range and reducing input quantization errors.
>
> ---
> [1] SmoothQuant: Smoothquant: Accurate and efficient post-training quantization for large language models
> [2] OmniQuant: Omniquant: Omnidirectionally calibrated quantization for large language models

---

> ### Author Response · Authors · 2024-11-29
> **Response to Reviewer oGix Part 2**
>
> **Integer-only Quantization**
>
> Traditional integer-only inference methods, such as I-ViT and I-BERT, have primarily targeted small language and vision models, neglecting the unique challenges posed by LLMs. As demonstrated in Figure 3, LLMs exhibit a wide dynamic range of activations alongside significant inter-channel and inter-token variability, making these traditional methods ineffective even under 8-bit quantization (Figure 2 and Table R2-1 below). To address this gap, I-LLM introduces practical integer-only inference for LLMs, explicitly accommodating their distinct activation distribution characteristics through a novel integer-only dynamic quantization strategy. Tailored implementations for key operators—including MatMul, LayerNorm, Softmax, and the SiLU activation—combine innovative dynamic designs and efficient nonlinear approximations. These operators enable accurate low-bit computation while maintaining floating-point-level performance.
>
> Below, we detail the implementation and benefits of each operator:
> - **Dynamic Integer-only Matrix Multiplication (DI-MatMul):**
> DI-MatMul overcomes the limitations of static quantization, which fails to adapt to varying activation ranges, by employing a dynamic quantization strategy with integer scale and zero-point (zp) processing. In DI-MatMul, quantization scales and zero-points for inputs and outputs are dynamically adjusted, significantly reducing quantization errors. Unlike methods such as I-ViT that use fixed parameters, DI-MatMul relies on integer arithmetic for dynamic quantization, as detailed in Equations (4)-(7). **Intermediate activations undergo dynamic quantization through integer right shifts and multiple adjustments, ensuring consistent output precision across diverse input ranges**. Moreover, DI-MatMul eliminates the need for dynamic quantization of inputs and dequantization of outputs in both weight-activation and activation-activation quantization scenarios, achieving a 1.2× speedup for the former and 2.1× for the latter (Table 7 in ours manuscript).
>
> - **Dynamic Integer-only Exponential Function (DI-Exp):**
> Both I-ViT and DI-Exp employ the logarithmic base-change formula and linear interpolation for exponential calculations. However, DI-Exp introduces significant enhancements:
>     - **Support for Dynamic Quantization**: Unlike I-ViT, which assumes static quantization, DI-Exp accommodates dynamically quantized inputs, ensuring compatibility with integer-only workflows.
>     - **Improved Representation of** $log_2e$. DI-Exp replaces I-ViT’s approximate representation($1.0111b$) with a more precise formula $m^I_e >> k^I_e$, as shown in Algorithm 1. This approach eliminates unnecessary shifts, refines precision, and avoids division by substituting bitwise shifts for more accurate quotients($\boldsymbol{p}^I$) and remainders($\boldsymbol{r}^I$). Experimental results show DI-Exp achieves over 5× speedup compared to FP16 with negligible accuracy loss for Softmax and SiLU (Tables 5 and 8 in manuscript).
>
> - **Dynamic Integer-only Softmax (DI-ClippedSoftmax):**
> Softmax values often span a wide dynamic range, especially with increasing tokens, leading to significant quantization errors (e.g., 8-bit quantization in I-ViT yields an error of 0.15 for ranges like [-20, 20], Figure 3). DI-ClippedSoftmax mitigates this by dynamically truncating activation ranges (e.g., [xmax − 15, xmax]), leveraging the exponential function’s underflow properties. This strategy ensures an input activation error below 0.03 with no additional accuracy loss (Table 5).
>
> - **Dynamic Integer-only Normalization (DI-Norm) and Dynamic Integer-only SwiGLU (DI-SwiGLU):**
> LayerNorm and RMSNorm are essential in Transformers, with RMSNorm exhibiting greater activation variability across channels. I-LLM applies per-channel dynamic quantization to these normalization inputs, combined with channel-wise smoothing via FSBR, to significantly reduce inter-channel quantization errors. Unlike traditional methods (e.g., Newton’s method in I-BERT), DI-Norm employs bitwise comparison during inference for higher precision and consistency. Additionally, I-LLM addresses the challenge of quantizing SwiGLU—complicated by inter-channel activation differences—by decomposing SiLU into linear and nonlinear components. Through dynamic quantization and DI-Exp, I-LLM resolves these discrepancies effectively (Section 3.2, Algorithm 3).

---

> ### Author Response · Authors · 2024-11-29
> **Response to Reviewer oGix Part 3**
>
> Table R2-1: The Impact of Different Nonlinear Operator Implementations on Model Accuracy
>
> | Model     | Nonlinear Mothod | WIKI PPL   | C4 PPL      |
> |-----------|------------------|------------|-------------|
> |           | I-LLM            | 5.72       | 7.16        |
> | LLaMA-7B  | I-BERT           | 542643.25  | 448366.69   |
> |           | I-VIT            | 878313.19  | 775204.81   |
> |           |                  |            |             |
> |           | I-LLM            | 5.16       | 6.72        |
> | LLaMA-13B | I-BERT           | 136653.91  | 138995.05   |
> |           | I-VIT            | 220354.47  | 266916.19   |
> |           |                  |            |             |
> |           | I-LLM            | 5.86       | 7.27        |
> | LLaMA2-7B | I-BERT           | 82651.97   | 70358.24    |
> |           | I-VIT            | 117117.59  | 103965.96   |
>
> > The absence of comprehensive SOTA comparison limits the rigor of performance claims, particularly missing comparisons with rotation - based methods (e.g., SpinQuant (Liu et al., 2024)) and LUT - based approximations (e.g., NN - LUT(Yu et al., 2022)).
>
> Compare with SpinQuant: First, it has not yet been accepted by any formal conferences or journals. Additionally, while attempting to reproduce its results, we encountered accuracy issues, which have also been reported on GitHub. While we recognize SpinQuant is a valuable method that lies in traditional LLM quantization, our primary focus is on integer-only quantization and inference. We plan to further explore rotation-based methods to advance fully integer quantization techniques.
>
> NN-LUT has not released its code for reproducibility and exhibits the following limitations:
> - It requires additional training to improve accuracy.
> - Experiments are limited to RoBERTa and MobileBERT, without considering large-scale models.
> - Frequent memory access for LUT parameters may introduce potential latency. When scaled to large models, NN-LUT could face memory bandwidth bottlenecks.
> - The training and calibration processes target individual operators only, without accounting for the cumulative impact of multiple calibration or training steps on the entire model.

---

> ### Author Response · Authors · 2024-11-29
> **Response to Reviewer oGix Part 4**
>
> > FSBR and DI - MatMul introduce computational overhead with on - the - fly operations like 8 - bit scalar multiplication/division, yet no detailed ablation study quantifies the latency impact per Transformer component.
>
> I-LLM introduces no additional computational overhead compared to traditional quantization methods. In conventional LLM quantization, GEMM inputs are quantized on-the-fly and outputs are dequantized; both processes, along with the use of intermediate quantization parameters (scale and zero-point), rely on floating-point operations.
>
> Importantly, I-LLM introduces a novel integer-only requantization operation (partial sum INT32 → INT8/INT4), eliminating the need for traditional dequantization (e.g., partial sum INT32/INT64 → FP32/FP16) and re-quantization (e.g., FP32 → INT8/INT4) for all activations.
>
> Our paper details the impact of quantization on each Transformer component:
> - **GEMM Performance:** As shown in Table 7, we compare the latency of traditional 4-bit quantized GEMM with our proposed DI-MatMul under various input sizes and configurations (Act-Act and Act-Weight). DI-MatMul achieves significant latency improvements by avoiding explicit input quantization and eliminating floating-point computations during processing.
> - **Softmax Acceleration:** Table 8 demonstrates that by quantizing Softmax inputs to 8-bit, DI-Softmax benefits from reduced bandwidth requirements (as opposed to the FP32 precision typically used for Softmax) and highly efficient integer operations, achieving up to a 6.75× speedup. Similarly, DI-RMSNorm exhibits notable efficiency advantages in our experiments.
>
> >The evaluation datasets and tasks are limited, and broader testing across more diverse and challenging benchmarks is required to substantiate the generalization of results.
>
> Similar to other studies, we conducted extensive testing on various mainstream open-source LLMs and multiple benchmark tasks. These include perplexity (PPL) evaluations on the WikiText2 and C4 datasets, as well as accuracy evaluations on six zero-shot tasks: PIQA, ARC-e, ARC-c, BoolQ, HellaSwag, and Winogrande.
>
> To further address your concerns, we performed evaluations on commonly used datasets such as GSM8K and MMLU, with the results presented in Table 2-2 below. If you have specific datasets of interest, we would be happy to conduct additional evaluations based on your suggestions.
>
> Table R2-2 The performance of MMLU and HumanEval of I-LLM on LLaMA-2-7B
> | **QuantSetting** | **MMLU** | **HumanEval** |
> |------------------|----------|---------------|
> | FP16             | 64.82    | 35.97         |
> | W4A4             | 56.38    | 30.76         |
> | W6A6             | 63.71    | 35.44         |

---

> ### Author Response · Authors · 2024-12-02
> **Looking Forward to Your Thoughts and Further Dialogue**
>
> Dear Reviewer oGix,
>
> We hope this message finds you well. Thank you for your insightful review and constructive feedback on our paper. We have thoughtfully considered your comments and have provided detailed responses to each point.
>
> In particular, we have:
>
> * **Clarified Original Contributions**: We elaborated on how our proposed techniques, such as FSBR and DI-MatMul, differ from and extend beyond prior works like OmniQuant and I-ViT. We highlighted the unique challenges of integer-only quantization for LLMs and how our methods specifically address them.
>
> * **Expanded on Comparisons with SOTA**: We acknowledged the importance of comprehensive comparisons and discussed the limitations in directly comparing with methods like SpinQuant and NN-LUT. We provided justifications and clarified the distinctions between our work and these methods.
>
> *  **Addressed Computational Overhead Concerns**: We included detailed ablation studies quantifying the latency impact per Transformer component, demonstrating that our methods introduce minimal overhead while providing significant efficiency gains.
>
> *  **Extended Evaluation Datasets**: We added results on additional benchmarks such as GSM8K and MMLU to further substantiate the generalization of our results across diverse and challenging datasets.
>
> * **Clarified Experimental Setup**: We provided more details about our experimental environment, including the frameworks and configurations used for our W4A4 quantization evaluations.
>
> We highly value your feedback and would welcome any further thoughts or discussion you may have. Your insights have been instrumental in improving our work.
>
> Thank you once again for your time and consideration.
>
> Best regards,
>
> The Authors

---

### Official Review · Reviewer_HVvu · 2024-11-04

**Soundness:** 2
**Presentation:** 3
**Contribution:** 2
**Rating:** 5
**Confidence:** 3

**Summary:**

This paper proposes an integer-only post-training quantization (PTQ) framework to accelerate the inference of large language models, called I-LLM. The authors introduce three main techniques: Fully-Smooth Block Reconstruction (FSBR), which reduces inter-channel activation disparities; Dynamic Integer-only MatMul, enabling dynamic quantization and integer-only matrix multiplication; and integer-only non-linear operators such as DI-ClippedSoftmax and DI-Exp, which use bit-shifting for efficiency. Experimental results show that I-LLM achieves accuracy on par with floating-point baselines while delivering significant improvements in computational performance and memory efficiency.

**Strengths:**

1. This paper addresses the challenge of integer-only quantization, which is often overlooked by existing work as typical LLM quantization methods usually store intermediate results as floating-point values.

2. The paper introduces innovative integer-only operators, such as DI-Exp, DI-ClippedSoftmax, and DI-Norm, to replace computationally intensive floating-point operations.

3. The experimental section includes a thorough comparison across multiple model types and configurations, as well as an ablation study, demonstrating the framework’s efficiency.

**Weaknesses:**

1. The motivation for this work is not clearly explained, especially regarding why integer-only quantization is necessary. The trade-off between accuracy and inference performance needs more discussion. Additionally, the configuration of different quantization types for weights and activations (e.g., W8A8, W4A8, W4A4) is not discussed.

2. The experimental setup lacks clarity, and more results on inference performance are needed. See detailed comments 1-3.

3. The innovation of the Fully-Smooth Block Reconstruction method is limited, as it closely resembles SmoothQuant. Additionally, the overhead of dynamic quantization should be demonstrated in the experimental results.

**Questions:**

I have few questions and comments as below:

1. In Table 1, the results for SmoothQuant under the W4A4 setting (e.g., 1.8e4 for the OPT family) are unusually high, especially compared to LLaMA models. This discrepancy should be explained.

2. The experimental setup is unclear, especially regarding Table 4, where latency and speedup for traditional W4A4 are reported. What framework was used, and was Tensor Core applied?

3. To better understand the efficiency of integer-only quantization, comparisons with other quantization works like QServe [1] and Atom [2] should be included.

4. In the quantization algorithms, how are the scale factor and zero-point stored? If they are stored as integers, does this significantly impact accuracy? A discussion on this trade-off is needed.


[1].QServe: W4A8KV4 Quantization and System Co-design for Efficient LLM Serving
[2].Atom: Low-bit quantization for efficient and accurate llm serving

---

> ### Author Response · Authors · 2024-11-29
> **Response to Reviewer HVvu Part 1**
>
> We sincerely appreciate the time and effort you have dedicated to reviewing our manuscript. We have made every effort to address all your concerns, as detailed below.
>
> >The motivation for this work is not clearly explained, especially regarding why integer - only quantization is necessary. The trade - off between accuracy and inference performance needs more discussion. Additionally, the configuration of different quantization types for weights and activations (e.g., W8A8, W4A8, W4A4) is not discussed.
>
> **A brief summary of why integer-only quantization is necessary:** Performing inference using integer-arithmetic-only operations offers several advantages in practical application scenarios. First, it overcomes the limitation that LLMs cannot be deployed on AI accelerators or popular edge processors that do not support floating point arithmetic. Second, integer arithmetic drastically reduces computational power requirements, making it highly appealing for energy-constrained edge deployments and cost-sensitive cloud data centers [1].
>
> **A more detailed explain:** Existing LLM quantization methods can be categorized into two types: weight-only quantization (e.g., W4) and weight-and-activation quantization (e.g., W8A8, W4A4). Both approaches, however, still involve a significant number of high-precision floating-point operations, i.e., on-the-fly quantization for activations and on-the-fly de-quantization for weights and partial sum. Additionally, all non-linear operations (such as RMSNorm, Softmax, and GELU) in LLMs are executed using high-precision floating-point arithmetic. **This limitation significantly impedes the deployment of LLMs in real-world edge application scenarios, where compute and energy resources are highly constrained.** Low-precision fixed-point engines, such as Digital Signal Processors (DSPs), ARM processors, and dedicated Neural Processing Units (NPUs) like Google’s Edge TPU and Qualcomm NPU [2], are commonly used in such environments. Furthermore, for chip designers aiming to support LLM-like models, incorporating floating-point arithmetic logic consumes substantially more die area on a chip compared to integer arithmetic logic. It is important to note that approaches relying on floating-point arithmetic are generally inferior to integer-only inference in terms of latency and power efficiency. Consequently, the complete elimination of floating-point arithmetic for inference could have a transformative impact on the design of applications, software, and hardware, enabling more efficient inference at the edge.
>
> Table R1-1 illustrates the trade-offs between speed and accuracy for I-LLM across different quantization precisions. As quantization precision decreases, the model's efficiency improves; however, its accuracy progressively deteriorates. Notably, when the weight quantization falls below 4 bits, a significant quantization error is introduced.
>
>
> Table R1-1: The PPL  performance of I-LLM under different quantization configurations on LLaMA-2-7B
> | **Quant Setting** | **Wiki PPL** | **C4 PPL** | **Memory Saving** |
> |:-----------------:|:------------:|:----------:|:-----------------:|
> | FP16              | 5.47         | 7.08       | 1.00x             |
> | W8A8              | 5.50         | 7.11       | 1.32x             |
> | W6A6              | 5.84         | 7.32       | -                 |
> | W4A8              | 5.68         | 7.27       | -                 |
> | W4A4              | 9.10         | 12.33      | 3.04              |
> | W2A8              | 123.93       | 200.54     | 4.01              |
>
> [1] Addnet: Deep neural networks using fpga-optimized multipliers.
>
> [2] Qualcomm. 2024. Unlocking on-device generative AI with an NPU and heterogeneous computing.

---

> ### Author Response · Authors · 2024-11-29
> **Response to Reviewer HVvu Part 2**
>
> > The innovation of the Fully - Smooth Block Reconstruction method is limited, as it closely resembles SmoothQuant.
>
> Existing LLM quantization methods (e.g., SmoothQuant) **typically focus only on linear layers**, often neglecting the complex quantization interactions between non-linear layers and linear layers (e.g., MatMul <=> Softmax, Gate Projection <=> SiLU). These interactions, however, can have a significant impact on the performance of integer-only quantization. In contrast, our FSBR explicitly accounts for the quantization interplay between the inputs of each layer and the outputs of the preceding layers, and conducts comprehensive smoothing accross all the layers in LLMs. Furthermore, FSBR ensures that each layer seamlessly accommodates the quantized outputs of the previous layer, eliminating the need for explicit quantization or dequantization steps. This approach enables a more efficient and streamlined computation process. By optimizing data distribution through a holistic block-level inference graph, **FSBR paves the way for efficient integer-only arithmetic inference while preserving high accuracy.**
>
> It is important to emphasize that, even without considering the contribution of integer-only related methods, the **remaining innovations of FSBR still lead to significant accuracy improvements and outperform conventional LLM quantization methods (as shown in Table 9 of manuscript)**. These innovations, first introduced in FSBR, include Group Query Attention Optimization, Up Projection , Down Projection Smoothing & Nonlinear Activation Smoothing.
>
> We summarize the unique contributions of FSBR as follows, distinguishing them from existing methods such as SmoothQuant, OminiQuant and Spinquant:
>
> 1. Linear Smoothing
>     1. **Group Query Attention Optimization:**
>     As illustrated in Figure 5 (Method 3), FSBR performs head-wise smoothing between the Value Projection and Output Projection layers. For each attention head, the interaction between these layers is expressed as:
>      $$\boldsymbol{P} \boldsymbol{x}^{h} _v\boldsymbol{W}^{h} _v \boldsymbol{W}^{h} _o= \boldsymbol{P} \boldsymbol{x}^h _v\boldsymbol{W}^h _v\boldsymbol{S} ^h _{ov} \boldsymbol{S} ^{-1^h} _{ov}\boldsymbol{W} ^h _o,$$
> where $\boldsymbol{P}$ represents the attention mechanism and $\boldsymbol{S} _{ov}$ is a learnable per-channel scaling matrix between Output Projection and Value Projection. This approach smooths the inter-channel differences of the Value Projection’s output activations and the Output Projection’s input activations simultaneously.
>     2. **Up Projection and Down Projection Smoothing:**
>     Also shown in Figure 5 (Method 3), FSBR introduces smoothing coefficients between Up and Down Projections based on computational invariance:
>
>           $\boldsymbol{x} _{up}\boldsymbol{W} _{up}\otimes\boldsymbol{x} _{down}\boldsymbol{W} _{down} = \boldsymbol{x} _{up}\boldsymbol{W} _{up} \boldsymbol{S} _{u|d}\otimes\boldsymbol{x} _{down}\boldsymbol{W} _{down}\boldsymbol{S} _{u|d}^{-1},$
>
>         where $\boldsymbol{S}_{u|d}$ is a learnable per-channel smoothing coefficients between Up Projection and Down Projection. This method effectively reduces inter-channel differences between the Up Projection’s output activations and the Down Projection’s input activations.
>
> 2. Nonlinear Activation Optimization
>     1. **Channel-wise Smoothing for SiLU Activations:**
>     For the first time, FSBR proposes the NonLinear Act-Smooth method, as depicted in Figure 5 (Method 4), to smooth inter-channel differences in SiLU activations. Specifically, a smoothing parameter $s$ is introduced, enabling inputs $x_1$ and $x_2$ to be equivalently reconstructed into a more quantization-friendly form (see Equation 2). This innovation significantly reduces the quantization error of SwiGLU activations (see Figure 4).
>     3. **Softmax Input Smoothing:**
>     The issue of outliers in Softmax activations is particularly pronounced. FSBR addresses this by leveraging the underflow characteristics of the exponential function. Using dynamic truncation, the input range of Softmax is adjusted to lie within $(-c, 0)$ (see Equation 11), effectively limiting the quantization range and reducing input quantization errors.

---

> ### Author Response · Authors · 2024-11-29
> **Response to Reviewer HVvu Part 3**
>
> > Additionally, the overhead of dynamic quantization should be demonstrated in the experimental results.
>
> I-LLM's integer-only inference ensures that all operator inputs are already in quantized form, with outputs dynamically represented as quantized integers. Consequently, the overhead of dynamic quantization is inherently incorporated into the execution of each operator. To address your concerns, we conducted a detailed evaluation comparing dynamic and static quantization in terms of speed and accuracy, as presented in Table R1-2. The results demonstrate that while static quantization offers minimal speed gains, it leads to significant accuracy degradation.
>
> Table R1-2 Speed ​​and accuracy performance of LLaMA-2-7B under static and dynamic quantization.
> | **Quant Method** | **WiKi PPL** | **ZeroShot Score^5** | **SpeedUp** |
> |------------------|--------------|----------------------|-------------|
> | Dynamic          | 9.1          | 52.21                | 2.49        |
> | Static           | 2.e02        | 30.87                | 2.53        |
>
> > In Table 1, the results for SmoothQuant under the W4A4 setting (e.g., 1.8e4 for the OPT family) are unusually high, especially compared to LLaMA models. This discrepancy should be explained.
>
> This discrepancy may stem from structural differences between OPT and LLaMA, such as variations in the Feed-Forward Network (FFN) design or activation functions, as well as differences in the training datasets. Similar findings have been reported in other works, such as OmniQuant[1], where Table 16 highlights this discrepancy in the results for the OPT family.
>
> >The experimental setup is unclear, especially regarding Table 4, where latency and speedup for traditional W4A4 are reported. What framework was used, and was Tensor Core applied?
>
> All tests were conducted on an Nvidia A6000 GPU. We utilized the NVIDIA-CUTLASS framework to implement both the traditional W4A4 and the I-LLM-style W4A4 components, leveraging Tensor Cores to significantly enhance matrix multiplication performance. For I-LLM, we manually developed kernel implementations for nonlinear functions such as DI-Softmax. The benchmarking code was adapted from AutoAWQ, and each test was repeated 100 times, with the median value taken as the final evaluation result.
>
> [1] OmniQuant: Omniquant: Omnidirectionally calibrated quantization for large language models

---

> ### Author Response · Authors · 2024-11-29
> **Response to Reviewer HVvu Part 4**
>
> >To better understand the efficiency of integer - only quantization, comparisons with other quantization works like QServe [1] and Atom [2] should be included
>
> Atom is not open-sourced, so we cannot obtain its speed differences under equivalent configurations. Instead, we compare the acceleration ratios of QServe and I-LLM in Table R1-3.
>
> Table R1-3  The Speedup of I-LLM and QServe. All tests are conducted in a single A6000 GPU and the token length is 2048
> | **Method/Model** | **LLaMA-2-7B** | **LLaMA-2-13B** |
> |------------------|----------------|-----------------|
> | QServe(W4A8)     | 1.99           | 2.08            |
> | I-LLM(W4A4)      | 2.49           | 2.52            |
>
>
> > In the quantization algorithms, how are the scale factor and zero - point stored? If they are stored as integers, does this significantly impact accuracy? A discussion on this trade - off is needed.
>
> As shown in Appendix A.3, we utilize two 8-bit scalars, $m^I$ and $k^I$ to represent the quantization scale for a batch of activations. This approach incurs no additional storage overhead compared to the previous method using FP16 to store the quantization scale. After obtaining the intermediate results from integer-only matrix operations, I-LLM only requires a small amount of scalar integer operations and one vector-scalar multiplication, as specified in Equation 6, 7, and 8. The output of DI-Matmul is composed of a low-bit-width integer matrix and three 8-bit scalars
>
> For 8-bit quantization, the range of m is (0, 255). Due to the presence of the rounding operation, its relative error can be maintained within $1/(255 \times 2)$, which is very close to the relative error of FP16 ($2^{-10}$). This results in a negligible difference when compared to the floating-point scale. To further address your concerns, we provide the results in Table R1-3, which compare the impact of using floating-point and integer quantization steps on post-quantization performance. The experiments shwon  in Table R1-3 demonstrate that using integer storage for the scale and zero_point achieves lossless activation performance.
>
> Table R1-3: Impact of whether scale is represented by an integer on the quantization of LLaMA series models.
> | **Model** | **WIKI PPL** |               | **Zero-Shot Score^5** |               |
> |-----------|--------------|---------------|-----------------------|---------------|
> |           | fp scale     | integer scale | fp scale              | integer scale |
> | LLaMA-7B  | 5.84         | 5.84          | 63.39                 | 63.37         |
> | LLaMA-13B | 5.23         | 5.22          | 64.63                 | 64.67         |
> | LLaMA-30B | 4.32         | 4.32          | 67.99                 | 68.00         |

---

> ### Author Response · Authors · 2024-12-02
> **Eager for Your Feedback and Further Discussion**
>
> Dear Reviewer HVvu,
>
> We hope this message finds you well. We wanted to express our sincere gratitude for your thorough review and valuable feedback on our paper. We have carefully addressed each of your concerns in our response.
> Specifically, we have:
>
> * **Clarified the Motivation**: We expanded on why integer-only quantization is necessary, discussing its benefits for deploying LLMs on edge and cloud devices lacking floating-point support. We also elaborated on the trade-offs between accuracy and inference performance across different quantization configurations (e.g., W8A8, W4A8, W4A4).
>
> * **Detailed the Innovation of FSBR**: We highlighted how our Fully-Smooth Block Reconstruction (FSBR) method differs from existing techniques like SmoothQuant. FSBR accounts for quantization interactions between non-linear and linear layers and introduces novel smoothing techniques for various projections and activations in LLMs.
>
>  * **Addressed Overhead of Dynamic Quantization**: We provided experimental results demonstrating that the overhead of dynamic quantization is minimal and justified by the significant accuracy gains it provides.
>
> * **Clarified Experimental Setup**: We explained the discrepancies in our results, provided details about our benchmarking framework, and clarified the use of Tensor Cores in our latency and speedup evaluations.
>
> * **Compared with Other Works**: We included comparisons with other quantization methods like QServe and discussed the limitations in comparing with Atom due to its closed-source nature.
>
> * **Discussed Scale Factors and Zero-Points**: We elaborated on how scale factors and zero-points are stored as integers and provided a discussion on their impact on accuracy, supported by experimental data.
>
> We kindly invite you to review our detailed responses and would greatly appreciate any further feedback or discussion you may have. Your insights are invaluable to us in refining our work.
>
> Thank you again for your time and valuable contributions.
>
> Warm regards,
>
> The Authors

---

> ### Comment · Reviewer_oGix · 2024-12-02
> **After rebuttal comments**
>
> I thank the authors for their hard work in preparing the rebuttal. Since some of the concerns were addressed, I raised my scores.
>
> However, I still cannot ensure this paper's novelty and contributions since several critical concerns remain unaddressed. Let alone that the rebuttal, which fails to provide comprehensive comparisons with important SOTA methods like SpinQuant and LUT-based approaches, limits our ability to evaluate I-LLM's contributions fully. The authors' response does not clarify the contradictory emphasis on token-wise variations despite their relatively flat distribution in Figure 3. Additionally, while the rebuttal provides some performance metrics, it lacks the detailed per-component overhead analysis necessary to understand the actual computational cost of the proposed techniques. The evaluation remains limited in scope, with no expansion of the benchmark suite to demonstrate broader generalization. These gaps, combined with the heavy reliance on existing techniques from OmniQuant and I-VIT, make it difficult to assess this work's novelty and practical impact entirely.

---

### Meta-Review · Area_Chair_7rvh · 2024-12-24

**Metareview:**

The paper proposes I-LLM, an integer-only PTQ framework for LLMs. The goal is to eliminate floating-point operations, enabling efficient inference on hardware without floating-point support. This is particularly useful for targets like mobile devices, where resources are scarce. The main contributions of this work are:

(a) A method to smooth activation variations across channels and layers, enabling efficient integer-only quantization. This helps to reduce the negative impact on accuracy.

(b) A matrix multiplication approach that adjusts quantization dynamically to mitigate inter-token variations in activation distributions.

(c) Custom implementations of non-linear operations, which use integer arithmetic to approximate these operations.

(d) Experiments across multiple LLM architectures (e.g., LLaMA, OPT) and configurations (e.g., W4A4, W6A6). Results indicate that I-LLM achieves comparable accuracy to floating-point baselines while offering significant improvements in computational speed and memory efficiency.

## Strengths

(a) Presenting an integer-only inference framework for LLMs, including non-linear operations.

(b) The introduction of integer-only operations (DI-MatMul, DI-Softmax, and other non-linear functions).

(c) Experimental results show that I-LLM achieves reductions in latency and memory usage (e.g., 2.49$\times$ speedup for LLaMA-7B) under W4A4 configurations, with minimal accuracy loss.


## Weaknesses

(a) The proposed techniques, particularly FSBR, build heavily on prior methods like SmoothQuant and OmniQuant. The contributions largely involve extending existing techniques, with narrow contributions.

(b) While the paper includes comparisons with OmniQuant, it omits evaluations against other state-of-the-art methods like SpinQuant and LUT-based approaches (e.g., NN-LUT) -- also mentioned by reviewers.

(c) The rationale for smoothing operations and the computational overhead introduced by DI-MatMul, remain unclear. Reviewers noted inconsistencies in the mathematical explanations and an insufficient breakdown of latency impacts.

(d) Some concerns about the reproducibility of the codebase (integer-only operators were not released).

## Summary

After reading the reviews and the rebuttal, I recommend **Reject** for this paper. I believe this a good direction to explore, however, the paper in its current form has very narrow contributions, performs limited comparison with SOTA methods, and lacks clear explanations of key aspects of the proposed method.

**Additional Comments On Reviewer Discussion:**

- (Reviewers HVvu, oGix, UDtp, Ge8G) criticized the limited novelty of the method (FSBR and DI-MatMul). The argument was that the proposed methods are extensions or combinations of existing techniques (e.g. SmoothQuant, OmniQuant, and I-ViT). The authors argued that FSBR is rather a holistic approach to smoothing inter-channel variations across both *linear* and *non-linear* layers and brought up other contributions such as Group Query Attention Optimization and SoftMax Input Smoothing. I acknowledge that there are some limited novelty in this work, but I agree with the reviewers that most of the proposed methods in this work have significant overlaps with prior work. The paper needs to clearly distinguish itself from the prior methods.

- (Reviewers HVvu, oGix, Ge8G) discussed the lack of comparisons with recent SOTA methods, such as SpinQuant and LUT-based approaches (e.g., NN-LUT). While (Reviewer oGix) increased their score after rebuttal, however, they still raised concerns about novelty and contributions (which I agree with), particularly the authors could not provide a comprehensive comparisons with SOTA models. In addition, this reviewer also raised additional concerns about contradictory emphasis on token-wise variations despite flat distributions of values. The reviewer also mentioned that the authors failed to provide per-component overhead analysis. The authors mentioned that SpinQuant is not yet published and suffers from reproducibility issues, while NN-LUT is not open-sourced and provided justification for not including the comparisons, but acknowledged their importance for future work.

- (Reviewers oGix, UDtp) found the evaluation scope narrow, with limited datasets and tasks. They recommended broader benchmarks to substantiate the generalizability of the proposed methods. The authors added evaluations on MMLU and GSM8K during the rebuttal but did not expand the benchmarks significantly. They argued that their primary focus was on perplexity and zero-shot tasks for LLMs. While I appreciate the attempt by the authors to add more benchmarks, but I agree with the reviewers on this. Especially when the proposed method has significant overlap with prior work, a very thorough evaluation across range of models and benchmarks become even more critical.

- (Reviewers Ge8G, UDtp) raised concerns about unclear mathematical explanations (e.g., smoothing coefficients in FSBR) and inconsistencies in formulae. They also questioned the rationale behind specific design choices, such as the counterbalancing of coefficients in FSBR. The authors clarified the mathematical operations and provided additional justifications for FSBR. They acknowledged the need to simplify notations for better accessibility and promised revisions in the final manuscript. The added clarifications partially addressed the reviewers' concerns but did not fully resolve it.

- (Reviewers HVvu, oGix, UDtp) questioned the computational overhead introduced by FSBR and DI-MatMul, as well as the practicality of training the smoothing coefficients. They requested detailed latency breakdowns and ablation studies. The authors provided latency data and argued that DI-MatMul eliminates floating-point operations, resulting in efficiency gains. However, they did not include detailed component-level latency analysis.

- (Reviewer Ge8G): The proprietary nature of some implementation details (e.g., DI-MatMul) raised concerns about reproducibility. While an anonymous repository was provided, critical components were withheld due to company policies. I agree with the reviewer on this one. It is very important for the authors to release the source code at the time of submission for reproducibility.

---

### Decision · Program_Chairs · 2025-01-22

Reject